# Progressive Alignment for Robust Domain Adaptation

## Abstract

Unsupervised Domain Adaptation (UDA) has advanced knowledge transfer between labeled source and unlabeled target domains, yet existing methods fall short in real-world scenarios where adversarial attacks threaten model reliability. Robustness against such attacks is essential but remains critically underexplored in UDA. Existing methods often treat domain alignment and adversarial defense as separate steps, causing unstable training, noisy pseudo-labels, and incomplete feature alignment ultimately limiting their effectiveness. Addressing both domain shift and adversarial robustness simultaneously is vital for deploying trustworthy models in dynamic, adversarial environments. In this work, we propose a robust UDA method from the perspective of multi-source and multi-target domain adaptation, treating clean and adversarial samples across both source and target as distinct domains. We aim to align both clean and adversarial domains across source and target within the adaptation framework. Therefore, we use progressive domain alignment strategy that explicitly aligns clean target features with multi-source domains through classifier discrepancy minimization, and implicitly aligns adversarial target features by enforcing classifier agreement on pseudo-labels. We find that this strategy effectively handles both domain shift and adversarial perturbations, leading to improved generalization and robustness. We demonstrate the effectiveness of our approach through extensive experiments on four benchmark datasets, accompanied by component-wise ablations. Our method achieves standard accuracies of 62.0%, 88.4%, 82.5%, and 73.7% and the corresponding robust accuracies under PGD-20 attack with $\epsilon = 2/255$ are 49.4%, 78.3%, 77.3%, and 72.1% on the *Office-Home*, *PACS*, *VisDA*, and *Digit* benchmark datasets, respectively.

## 1 Introduction

Unsupervised domain adaptation (UDA) aim to transfer knowledge from a labeled source domain to an unlabeled target domain under a distribution shift. Popular UDA methods (Ganin et al., 2016a; Long et al., 2018a; 2017; Saito et al., 2018a; Sun & Saenko, 2016a; Xu et al., 2019; Choi et al., 2019; Yang et al., 2021a) focus on aligning features between the source and target domains to reduce the domain gap. These alignments are typically achieved using statistical discrepancy minimization schemes, (Ganin et al., 2016b; Baktashmotlagh et al., 2016; Gong et al., 2012; Pan et al., 2010; Peng et al., 2019; Sun & Saenko, 2016b;a), adversarial training objectives (Ganin et al., 2016a; Long et al., 2018a;b; Tzeng et al., 2017; Xu et al., 2018; Zhao et al., 2018) , or domain-specific transformations (Chang et al., 2019; Li et al., 2018; Roy et al., 2019). While these methods perform well in standard domain shift assumptions, they often fail in more realistic scenarios where data may be perturbed or adversarially attacked. Such perturbations can significantly degrade the performance of adapted models (Mehra et al., 2021; Shi & Liu, 2023), especially in downstream tasks (e.g., classification). As a result, despite their progress, conventional UDA methods still face the open challenge of ensuring robustness against adversarial perturbations, a problem made particularly difficult by domain shift and the missing target labels.

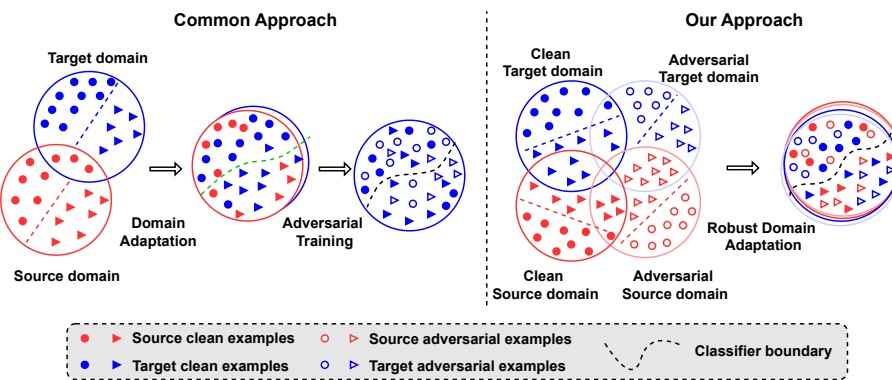

Figure 1: (Best viewed in color.) Comparison of approaches between SOTA and Ours for adversarially robust domain adaptation. **Left**: SOTA approach first perform source–target distribution alignment using standard UDA technique, followed by adversarial training on the target domain. This results in suboptimal performance due to misalignment between clean and adversarial domains and noisy pseudo-labels. **Right**: Our proposed approach explicitly treats clean source, adversarial source, clean target, and adversarial target as distinct domains, aligning their distributions(as detailed in Fig 2). This allows adversarial robustness to be integrated as a core component of the domain adaptation process.

Recently, several robust UDA methods (Awais et al., 2021; Wang et al., 2024a; Lo & Patel, 2022; Zhu et al., 2023; Yang et al., 2021b; Soni & Dutta, 2025) have attempted to address the dual challenge of domain shift and robustness to input perturbations. The most common strategy first performs domain adaptation from the source to the target domain, followed by pseudo-label generation and then perform adversarial training on the target domain (as illustrated in left of Figure 1). Existing methods (Wang et al., 2024a; Zhu et al., 2023; Soni & Dutta, 2025) primarily focus on refining noisy pseudo-labels in the target domain and leveraging adversarial examples during training to improve model robustness in the later training stages. However, this decoupling causes three issues: (i) noisy pseudo-labels induce confirmation bias (Han et al., 2020) (ii) decision boundaries learned via adversarial training remain misaligned across domains, and (iii) models tend to overfit both clean and adversarial target samples. It leads to unstable learning and ultimately poor generalization to the target domain under both domain shift and adversarial perturbations. Other approaches (Awais et al., 2021; Lo & Patel, 2022) utilize robust pretrained models or self-supervised signals for improving robustness against input perturbation. However, their gains remain limited compared to conventional adversarial training, where ground truth provides strong supervision, and pretrained models are not always available in practice. One problem shared among all existing robust UDA methods is that they treat adversarial robustness as an auxiliary task decoupled from domain adaptation. Consequently, clean and adversarial distributions across domains remain misaligned in the latent space, fundamentally limiting robust generalization. See Figure 1 for a simple illustrative example.

In this work, we propose (to the best of our knowledge) the first robust UDA framework formulated as a multi-source, multi-target domain alignment problem, where we treat the clean source, adversarial source, clean target, and adversarial target as distinct domains. This formulation jointly captures both the cross-domain shift (source → target) and the clean–adversarial gap within the representation space. Although adversarial perturbations are imperceptible to humans, they induce a measurable distribution shift (Song et al., 2018; Liao et al., 2018) between clean and adversarial examples, motivating us to treat adversarial examples from both source and target as separate domains. Explicitly aligning all four domains in the latent space is essential for achieving robust generalization on the target domain under distribution shift and adversarial perturbations. To realize this formulation, we propose a progressive alignment method that gradually aligns clean and adversarial domains across source and target, thereby integrating robustness directly into the adaptation process. Our approach leverages a shared feature extractor with two classifiers and progressively stabilizes clean and adversarial source alignment, adapts clean targets, and transfers robustness to adversarial targets.

Figure 2: (Best viewed in color.) The architecture consists of a shared feature extractor and two classifier heads. We illustrate the framework in a two-class setting using both clean and adversarial samples from source and target domains. In (b), clean target samples initially lie outside the support of the multi-source distribution, resulting in classifier disagreement in the discrepancy region. We perform explicit alignment by minimizing this discrepancy, aligning clean target features with the source-induced latent space. In (c), implicit alignment is achieved by enforcing classifier agreement on pseudo-labeled adversarial target samples, encouraging their alignment toward class-consistent regions and improving robustness under domain shift.(*adv*: adversarial, *tgt*: target, *src*: source).

This design ensures stable adaptation while mitigating the pitfalls of prior auxiliary task approaches. The key intuition is that supervised alignment of clean and adversarial sources provides a stable latent space, while progressively incorporating clean and adversarial targets transfers robustness without destabilizing the learned representations. To summarize, our method demonstrates effective robust domain adaptation by progressively leveraging explicit and implicit alignment, enabled by two classifiers architecture. We conduct extensive evaluations on four benchmark datasets using the widely adopted ResNet-50 backbone (He et al., 2016), and derive key insights through comprehensive empirical analysis.

Our main contributions are summarized as follows:

• We propose, to the best of our knowledge, the first formulation of robust UDA as a *multi-source, multi-target domain alignment problem*, where clean source, adversarial source, clean target, and adversarial target are treated as distinct domains. This formulation explicitly captures both the cross-domain shift (source → target) and the clean–adversarial gap, rather than treating robustness as an auxiliary task.

• We propose a *progressive alignment strategy* that unfolds in three stages: (i) alignment of clean and adversarial sources under label supervision to stabilize the latent space, (ii) explicit alignment of clean target features via classifier discrepancy minimization, and (iii) implicit alignment of adversarial target features through double consistency filtering and curriculum learning.

• We introduce a principled dual-classifier architecture designed to align decision boundaries across clean and adversarial across domains, detect boundary-level misalignment through classifier disagreement, and provides a stable supervision signal for pseudo-labeling. This structure enables unified and reliable adaptation in both standard and adversarial target scenario.

• We conduct extensive experiments on four UDA benchmarks. Our method obtained standard accuracies of 62.0%, 88.4%, 82.5%, and 73.7% and the corresponding robust accuracies under PGD-20 attack with $\epsilon = 2/255$ are 49.4%, 78.3%, 77.3%, and 72.1% on the *Office-Home*, *PACS*, *VisDA*, and *Digit* benchmark datasets, respectively. We achieved stable adaptation and improved robust generalization on both clean accuracy and robustness against adversarial attacks.

## 2 METHOD

**Notations.** Let $\mathcal{X}$ and $\mathcal{Y}$ denote input and label spaces, respectively. We have access to two labeled source domain datasets: clean source $\mathcal{D}_s^{\text{cln}} = \{(x_s^i, y_s^i) \in \mathcal{X} \times \mathcal{Y}\}$ and adversarial source $\mathcal{D}_s^{\text{adv}} = \{(\hat{x}_s^i, y_s^i) \in \mathcal{X} \times \mathcal{Y}\}$, as well as two unlabeled target domain datasets: clean target $\mathcal{D}_t^{\text{cln}} = \{x_t^i \in \mathcal{X}\}$ and adversarial target $\mathcal{D}_t^{\text{adv}} = \{\hat{x}_t^i \in \mathcal{X}\}$. We emphasize that the adversarial source and adversarial target domains are not fixed datasets; instead, the adversarial examples $\hat{x}_s$ and $\hat{x}_t$ are generated during training iteration using Projected Gradient Descent Madry et al. (2017). All domains share a common label space $\mathcal{Y}$ with $K$ classes. We employ a deep neural network comprising a shared feature extractor $\mathcal{F} : \mathbb{R}^{224 \times 224 \times 3} \to \mathbb{R}^d$, and two classifier heads $\mathcal{H}_1, \mathcal{H}_2 : \mathbb{R}^d \to \mathbb{R}^K$, which are shared across domains. For compactness, we define $\mathcal{H}$ as the function whose output is the vertical concatenation of the logits produced by the two classifier heads $\mathcal{H}_1$ and $\mathcal{H}_2$. The network outputs are denoted by logits matrix $\mathbf{Z} = \mathcal{H} \circ \mathcal{F}(x) \in \mathbb{R}^{2 \times K}$, where $\circ$ denotes function composition. The matrix $\mathbf{Z}$ is obtained by stacking logits produced by $\mathcal{H}_1$ and $\mathcal{H}_2$ applied to shared feature representation $\mathcal{F}(x)$.

**Robustness Setup.** We adopt a general adversarial threat model where, for a given input $x \in \mathcal{X}$, the adversarial example $\tilde{x}$ is constrained within a perturbation set $\mathcal{B}(x) \subset \mathbb{R}^d$. A common formulation is the $\ell_p$-bounded perturbation set defined as $\mathcal{B}(x) = \{\hat{x} \mid \|\hat{x} - x\|_p \leq \epsilon\}$, where $\epsilon$ controls the perturbation strength. In our work, we use the $\ell_\infty$ norm and generate adversarial examples using Projected Gradient Descent(PGD) (Madry et al., 2017) under this constraint. Adversarial examples are generated using PGD with respect to the feature extractor $\mathcal{F}$ and both classifier heads $\mathcal{H}_1$ and $\mathcal{H}_2$.

### 2.1 WARM-START TRAINING

We aim to learn a good initial latent feature space and stable classifiers by training the model on each source domains. We refer to this as the warm-start process, which prepares the model for effective adaptation to the target domains. Our model has shared feature extractor $\mathcal{F}$ and two classifiers $\mathcal{H}_1$ and $\mathcal{H}_2$, which are shared across each source domains (Figure 2a). Each classifier is initialized independently with distinct random weights and is trained on the same set of source samples within each mini-batch. For a clean source-domain instance $x_s$, we obtain the output matrix $\mathbf{Z}_s^{\text{cln}} = \mathcal{H} \circ \mathcal{F}(x_s)$, where $\circ$ denotes function composition. The matrix $\mathbf{Z}_s^{\text{cln}}$ is constructed by stacking the logits from both classifiers. To define the class probability vector $p$, we compute a convex combination of the probabilities assigned by each classifier:

$$p = \frac{1}{2} \sum_{i=1}^{2} \sigma(\mathbf{Z}_s^{\text{cln}}[i \cdot]), \tag{1}$$

where $\sigma(\cdot)$ denotes softmax function applied to the logits from each classifier, and $\mathbf{Z}_s^{\text{cln}}[i \cdot]$ represents the $i$-th row of matrix $\mathbf{Z}_s^{\text{cln}}$ corresponding to classifier $\mathcal{H}_i$. Treating $p$ as the class probability vector, we minimize the standard cross-entropy loss with respect to the ground-truth clean source label $y_s$:

$$\mathcal{L}_{\text{CE}}(p, y_s) = -\log(p_{y_s}) = -\log\left(\frac{1}{2} \sum_{i=1}^{2} \sigma(\mathbf{Z}_s^{\text{cln}}[i \cdot])[y_s]\right). \tag{2}$$

Applying Jensen's inequality to the concave log function (Eq. 2), the above loss can be upper bounded by the average of individual classifier cross-entropy losses (Eq. 3), which simplifies the optimization and encourages consistent predictions across classifiers. Similarly, this process is applied to the adversarial source samples $\hat{x}_s$, whose logits are $\mathbf{Z}_s^{\text{adv}} = \mathcal{H} \circ \mathcal{F}(\hat{x}_s)$. Thus, the overall training objective during warm-up is:

$$\min_{\mathcal{F}, \mathcal{H}} \mathbb{E}_{\mathcal{D} \in \{\mathcal{D}_s^{\text{cln}}, \mathcal{D}_s^{\text{adv}}\}} \mathbb{E}_{(x_s, y_s) \in \mathcal{D}} \left[\frac{1}{2} \sum_{i=1}^{2} -\log\left(\sigma(\mathbf{Z}_s^{\mathcal{D}}[i, \cdot])[y_s]\right)\right], \tag{3}$$

where $\mathcal{D}_s^{\text{cln}}$ and $\mathcal{D}_s^{\text{adv}}$ denote the clean and adversarial source datasets, respectively. The objective in Eq. 3 is minimized using mini-batch stochastic optimization, where each mini-batch contains an equal number of samples from both source domains. This strategy gradually increases the similarity between the classifiers by encouraging agreement in their predicted class labels for source samples. This step provides a warm start, preparing the model for effective adaptation to the target domains in subsequent training stages.

## 2.2 Explicit Alignment of Clean Target domain

After the warm-start, we align the clean target domain with the shared latent space formed by the clean and adversarial source domains. Although the classifiers are trained on diverse source distributions, a domain gap persists between the clean target features and the multi-source features, as target samples are more likely to lie near class decision boundaries. Consequently, clean target samples that lie outside the support of the multi-source distributions are likely to induce disagreement between the classifiers, $\mathcal{H}_1$ and $\mathcal{H}_2$. This discrepancy region is illustrated on the top of the Figure 2b.

To address this misalignment, we train the shared feature extractor and classifiers in an adversarial manner (Saito et al., 2018b). We first train the classifiers ($\mathcal{H}_1$ and $\mathcal{H}_2$) to maximize the discrepancy between their predictions on clean target features, thereby identifying target samples that lie outside the support of the multi-source feature distribution. Subsequently, we update the shared feature extractor $\mathcal{F}$ to minimize this discrepancy, encouraging clean target samples to align with the multi-source latent space. This training process alternating between maximizing classifier disagreement and minimizing it through the feature extractor. During the adaptation process, we first fix the shared feature extractor and train the classifiers as follows:

$$\mathcal{L}_{\text{disc}} = \mathbb{E}_{x_t \in \mathcal{D}_t^{\text{cln}}} \left[ D\left( p_1(x_t), p_2(x_t) \right) \right], \quad \max_{\mathcal{H}_1, \mathcal{H}_2} \mathcal{L}_{\text{disc}}. \tag{4}$$

To retain source knowledge, we also minimize the cross-entropy loss on labeled source domains (Clean and Adversarial) during training (as defined in equation 3). Next, fix the classifiers and train the feature extractor to minimize classifier discrepancy, encouraging clean target features to move closer to shared latent space:

$$\min_{\mathcal{F}} \mathcal{L}_{\text{disc}}, \tag{5}$$

where $\mathcal{D}_t^{\text{cln}}$ denotes the clean target domain. We quantify the classifier discrepancy using the $L_1$ distance between their softmax outputs:

$$D(p_1, p_2) = \sum_{k=1}^{K} |p_1(k) - p_2(k)|, \tag{6}$$

where $p_1(k)$ and $p_2(k)$ denote the predicted probabilities for class $k$ from $\mathcal{H}_1$ and $\mathcal{H}_2$, respectively. Following prior work Saito et al. (2018a), we adopt the $L_1$ distance, as it provides sharper sensitivity to class-wise prediction disagreements compared to $L_2$, facilitating more effective detection of target samples near decision boundaries. This training process not only simplifies the adaptation strategy but also enables clean target samples to align naturally within the shared latent space, leveraging the rich and diverse representations learned from multiple source distributions.

Determining Convergence with Target Consistency Rate. We determine the convergence of the target alignment process by tracking the consistency of classifier predictions on clean target samples, instead of monitoring the discrepancy loss as done in prior work Saito et al. (2018b). This consistency-based criterion offers a more direct and label-free signal for target adaptation and deciding when to transition to the self-training phase. We define the consistency indicator for a target sample $x_t$ as:

$$c(x_t) = \begin{cases} 1, & \text{if } \arg\max \mathbf{Z}_t^{\text{cln}}[1, \cdot] = \arg\max \mathbf{Z}_t^{\text{cln}}[2, \cdot], \\ 0, & \text{otherwise}, \end{cases} \tag{7}$$

where $\mathbf{Z}_t^{\text{cln}}[1, \cdot]$ and $\mathbf{Z}_t^{\text{cln}}[2, \cdot]$ denotes the outputs of classifiers $\mathcal{H}_1$ and $\mathcal{H}_2$, respectively. The overall Target Consistency Rate over the unlabeled clean target set $\mathcal{D}_t^{\text{cln}}$ is given by:

$$\mathcal{C}(\mathcal{D}_t^{\text{cln}}) = \frac{1}{|\mathcal{D}_t^{\text{cln}}|} \sum_{x_t \in \mathcal{D}_t^{\text{cln}}} c(x_t). \tag{8}$$

We track $\mathcal{C}(\mathcal{D}_t^{\text{cln}})$ across training epochs. When this rate saturates (*i.e.,* stops increasing for several consecutive epochs), we consider the alignment stage converged (see in Sec. 3.3). This provides a reliable criterion for proceeding to the next phase of training without access to target labels.

## 2.3 Implicit alignment of Adversarial target Domain

After achieving explicit alignment, we introduce adversarial target samples into the adaptation process (Figure 2c). Since adversarial examples are generated with respect to class labels, we adopt a double consistency criterion to ensure pseudo-label reliability and mitigate confirmation bias Han et al. (2020). We

first obtain pseudo-labels for the clean target domain based on the predictions of both classifiers. Let $\mathcal{D}_t^{\text{cln}}$ denote the set of unlabeled clean target samples. We identify a subset of samples for which both classifiers agree on the predicted class:

$$\mathcal{D}_t^{(0)} = \left\{ (x_t, \hat{y}_t) \;\middle|\; x_t \in \mathcal{D}_t^{\text{cln}}, \; \arg\max \mathbf{Z}_t^{\text{cln}}[1, \cdot] = \arg\max \mathbf{Z}_t^{\text{cln}}[2, \cdot] \right\}, \tag{9}$$

where $\mathbf{Z}_t^{\text{cln}} = \mathcal{H} \circ \mathcal{F}(x_t)$ denotes the matrix of predicted logits from both classifiers for a target sample $x_t$. $\hat{y}_t$ is the agreed-upon pseudo-label. For each $x_t \in \mathcal{D}_t^{(0)}$, we generate an adversarial counterpart $\hat{x}_t$ based on agreed pseudo-label. To ensure reliability of pseudo-labeled adversarial samples, we retain only those adversarial samples where classifier consistency persists and predictions remain consistent with clean sample:

$$\arg\max \mathbf{Z}_t^{\text{adv}}[1, \cdot] = \arg\max \mathbf{Z}_t^{\text{adv}}[2, \cdot] = \hat{y}_t. \tag{10}$$

The resulting robust and consistent subset is denoted by $\mathcal{D}_t^{\text{adv}}$. For an adversarial target sample $\hat{x}_t$, we denote its logits as: $\mathbf{Z}_t^{\text{adv}} = \mathcal{H} \circ \mathcal{F}(\hat{x}_t)$ . Initially, training is performed on $\mathcal{D}_t^{\text{adv}}$, which contains only the most reliable and adversarially stable samples. As training progresses, we gradually expand the training set by including additional adversarial samples from $\mathcal{D}_t^{(0)} \setminus \mathcal{D}_t^{\text{adv}}$, ranked according to confidence score:

$$\text{confidence}(x_t) = \max_j \left( \frac{1}{2} \sum_{i=1}^{2} \sigma\left( \mathbf{Z}_t^{\text{cln}}[i, \cdot] \right)_j \right), \tag{11}$$

where $\mathbf{Z}_t^{\text{cln}}[i, \cdot]$ and $\sigma(\cdot)$ represent the output of classifier $\mathcal{H}_i$ for clean target sample and softmax function. This forms a curriculum learning strategy Xu et al. (2023) from easy (high-confidence) to hard (low-confidence) adversarial examples. The objective adaptation using these adversarial target samples is defined by:

$$\min_{\mathcal{F}, \mathcal{H}} \; \mathbb{E}_{(\hat{x}_t, \hat{y}_t) \in \mathcal{D}_t^{\text{adv}} \cup \mathcal{D}_t^{\text{curr}}} \left[ \frac{1}{2} \sum_{i=1}^{2} -\log\left( \sigma\left( \mathbf{Z}_t^{\text{adv}}[i, \cdot] \right) [\hat{y}_t] \right) \right], \tag{12}$$

where $\mathcal{D}_t^{\text{curr}} \subseteq \mathcal{D}_t^{(0)} \setminus \mathcal{D}_t^{\text{adv}}$ denotes the subset of pseudo-labeled clean target samples progressively added during training via a confidence-based curriculum schedule. Following this, the model is trained on both clean and adversarial source domains, as well as the pseudo-labeled clean and adversarial target samples, using the objectives defined in Eq. 12 and Eq. 3 using alternate mini-batch from both source and target domains. Note that Eq. 12 is also used for clean target samples by replacing the adversarial input with its clean counterpart. To maintain pseudo-label quality, we update the pseudo-labels every $n$ epochs using the latest classifier predictions on the clean target domain. Training continues until the classifier consistency rate on clean target samples converges, which serves as a proxy for overall convergence. This is based on the assumption that adversarial target samples, generated from consistent clean targets, become implicitly aligned through agreement-based training on their pseudo-labels. This strategy stabilizes self-training and improves robustness under adversarial perturbations.

## 3 EXPERIMENTS

We compare our method against standard UDA baselines and recent robust UDA approaches, including MDD (Zhang et al., 2019b), SDAT (Rangwani et al., 2022), LUHP (Zhang et al., 2024), DANN (Ganin et al., 2016a), AT (Madry et al., 2017), UDA+AT (Madry et al., 2017), UDA+TRADES (Zhang et al., 2019a), ARTUDA (Yang et al., 2021b), SRoUDA (Zhu et al., 2023), DART (Wang et al., 2024a), and CAM+SPLR (Soni & Dutta, 2025). Dataset and baseline descriptions are provided in section B of appendix.

**a) Implementation details.** We implement our method using PyTorch (Paszke et al., 2019), with the ResNet-50 architecture as the CNN backbone. The model is optimized using the Adam optimizer ()kingma2014adam with a learning rate of $10^{-4}$ and a weight decay of $5 \times 10^{-4}$. During training, Adversarial data are generated using 5 steps of Projected Gradient Descent (PGD), with a step size of $1/255$. The perturbations are constrained within an $\ell_\infty$-norm ball defined as $\mathcal{B}(x) = \{\hat{x} \mid \|\hat{x} - x\|_\infty \leq \epsilon\}$, where $\epsilon$ is set to $2/255$. During the warm-start, we optimize Eq.3 and use small, separate validation sets (10%) from both clean and adversarial source domains to determine early stopping. The objectives in Eq.4 and Eq.5 are optimized in an alternating fashion, while the target consistency rate (Eq.8) is periodically monitored to determine

convergence. Similarly, during self-training, Eq.3 and Eq.12 are also optimized in the same alternating fashion. We update the target pseudo-labels every $n = 10$ epochs, following the procedure outlined in Algorithm 1 (Section C, supplementary). The total number of training iterations is determined based on the convergence of the target consistency rate for each dataset. See the supplementary material (Section B) for architecture details and hyperparameters.

**b) Evaluation.** We evaluate our method on the target domain using natural accuracy for clean target data and robust accuracy for adversarial target data. Robust accuracy is computed under a 20-step PGD attack (PGD-20) with $\epsilon = \frac{2}{255}$ for all datasets, and $\epsilon = \frac{8}{255}$ for the Digit and PACS datasets. We report the mean and standard deviation over three independent runs for all experiments.

## 3.1 COMPARATIVE RESULTS

We present the results in Table 1, evaluated under a PGD-20 attack with $\epsilon = 2/255$ across all source–target pairs for each dataset: 12 pairs for Office-Home, 12 pairs for PACS, 2 pairs for VisDA, and 3 pairs for Digit. The prior baseline results are reported from (Wang et al., 2024a) and (Soni & Dutta, 2025).Due to page limits, detailed results for each individual source–target domain pair across all datasets are provided in the supplementary material (Section D). Table 1 shows that our method significantly improves adversarial robustness over existing baselines across all four benchmarks. In particular, it outperforms robust UDA methods such as ARTUDA and SROUDA, as well as source-only baselines like AT (src only), and target-only baselines (i.e., UDA+AT and UDA+TRADES) in terms of robust accuracy. Common UDA method collapses under PGD attack. While recent robust UDA methods such as DART and CAM+SPLR demonstrate competitive performance, our approach consistently achieves higher robustness, underscoring its effectiveness against both domain shift and adversarial perturbations. Results comparing performance under a PGD-20 attack with $\epsilon = 8/255$ on three domain pairs of the PACS dataset and two domain pairs of the Digit dataset are included in Section D.2. Additional results on source test data across datasets are presented in Section D.3. Also additional results on Auto-Attack (Croce & Hein, 2020) and FGSM are included with different backbones including ResNet-50, ResNet-101, and transformer-based backbones in Section F of the appendix.

Table 1: Natural accuracy (Nat.) and Robust accuracy under PGD attack (PGD) on target test data, averaged over all source–target domain pairs across Office-Home, PACS, VisDA, and Digit. [†] Not reported as mean ± standard deviation.

| Datasets | Office-Home | | PACS | | VisDA | | Digit | |
|---|---|---|---|---|---|---|---|---|
| **Method** | Nat | PGD | Nat. | PGD | Nat. | PGD | Nat. | PGD |
| MDD* | 68.1±0.0 | – | – | – | – | – | – | – |
| SDAT* | 72.2±0.0 | – | – | – | – | – | – | – |
| LUHP* | **75.4±0.0** | – | – | – | – | – | – | – |
| DANN | 57.4±0.2 | 1.5±0.1 | 81.1±0.3 | 11.0±0.2 | 73.0±0.4 | 6.0±0.1 | 72.9±0.3 | 68.5±0.3 |
| AT(src only) | 49.7±0.7 | 31.2±0.1 | 65.7±0.9 | 48.2±0.1 | 36.2±0.5 | 29.8±0.3 | **74.3±0.4** | 70.0±0.1 |
| UDA+AT | 52.1±0.2 | 40.5±0.2 | 82.0±0.4 | 70.0±0.2 | 77.7±0.2 | 70.3±0.2 | 73.5±0.2 | 71.5±0.3 |
| UDA+TRADES | 53.4±0.3 | 41.0±0.3 | 82.3±0.2 | 71.7±0.3 | 76.6±0.1 | 70.2±0.3 | 73.3±0.3 | 71.5±0.3 |
| ARTUDA | 70.3±0.3 | 37.1±0.6 | 74.6±0.6 | 67.4±0.5 | 58.9±1.6 | 56.2±1.3 | 72.7±0.1 | 72.1±0.2 |
| SROUDA | 51.3±0.2 | 38.7±0.6 | 72.7±0.1 | 64.0±0.5 | 64.7±1.9 | 53.2±1.0 | 73.5±0.6 | 71.7±0.1 |
| DART | 56.4±0.1 | 40.7±0.1 | 85.5±0.1 | 73.3±0.0 | 78.4±0.1 | 71.7±0.2 | 73.5±0.1 | 72.0±0.1 |
| CAM+SPLR[†] | 58.7 | 45.9 | 87.4 | 76.7 | 81.1 | 76.5 | – | – |
| **Ours** | 62.0±0.2 | **49.4±0.2** | **88.4±0.1** | **78.3±0.1** | **82.5±0.1** | **77.3±0.2** | 73.7 ± 0.1 | **72.1 ± 0.1** |

(*) Results available only for Office-Home.

## 3.2 COMPONENT ANALYSIS

We evaluate the contribution of each component in our method through an ablation study on two representative domain adaptation tasks from PACS and Office-Home. Specifically, we consider following configurations: (1) warm-start using only clean or adversarial source; (2) removal of explicit alignment using clean target; (3) removal of implicit alignment based on adversarial target examples; (4) disabling the double consistency criterion for selecting stable pseudo-labels; (5) fixing pseudo-labels throughout training without updates; and (6) removing curriculum scheduling by using all pseudo-labeled samples at once. The results, summarized

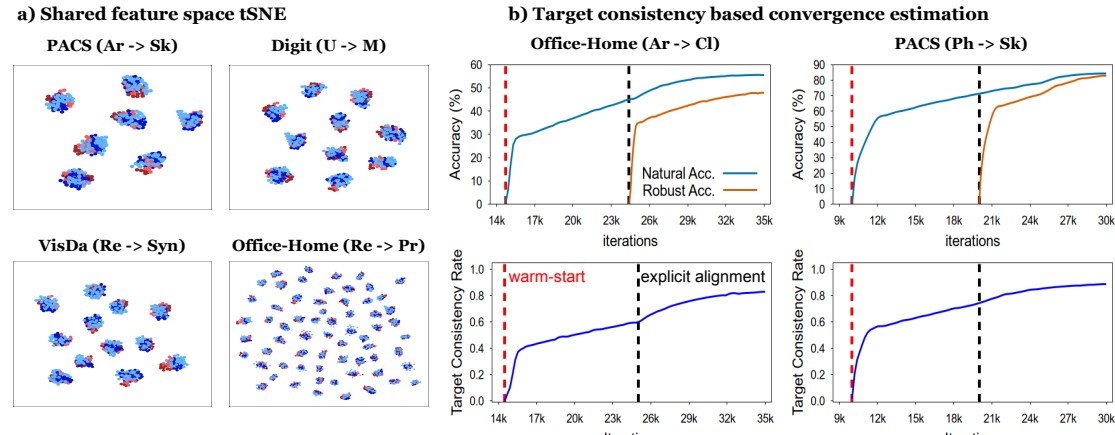

Figure 3: a) **t-SNE.** We visualize the alignment of domains across both clean and adversarial inputs using features extracted from the shared feature extractor (i.e., the output of $\mathcal{F}$). b) **Target consistency rate.** The target consistency rate (bottom) serves as an indicator of convergence behavior during explicit and implicit training (top). (Legend: • Clean Source, • Adversarial Source, • Clean Target, • Adversarial Target)

in Table 2, show that excluding either explicit or implicit alignment significantly reduces PGD robustness, underscoring the need to align both clean and adversarial domains. Using both clean and adversarial source data during warm-up yields a better performance in both natural and robust than using either alone. Further, eliminating double agreement or fixing pseudo-labels leads to performance degradation, confirming the importance of pseudo-label reliability and refinement. Finally, skipping curriculum scheduling harms performance, demonstrating the benefit of gradually incorporating samples based on confidence. Additional analyses and ablation studies are included in Section G of the Appendix.

Table 2: Component-wise ablation on two source–target pairs from PACS and Office-Home. We report Natural (Nat.) and robust (PGD) accuracy (%), evaluating the impact of each component on clean and adversarial performance.

| Source → Target | Sk → Ca | | Re → Pr | |
|---|---|---|---|---|
| Method | Nat. | PGD | Nat. | PGD |
| Warm-start w/ Clean Source only | $78.5 \pm 0.1$ | $67.3 \pm 0.2$ | $73.9 \pm 0.2$ | $62.6 \pm 0.3$ |
| Warm-start w/ Adv Source only | $76.8 \pm 0.1$ | $68.9 \pm 0.1$ | $72.5 \pm 0.2$ | $63.7 \pm 0.2$ |
| w/o Explicit Alignment | $73.9 \pm 0.1$ | $64.7 \pm 0.3$ | $70.7 \pm 0.2$ | $60.5 \pm 0.2$ |
| w/o Implicit Alignment | $74.4 \pm 0.2$ | $63.5 \pm 0.4$ | $71.8 \pm 0.2$ | $59.3 \pm 0.2$ |
| w/o Double Consistency | $79.8 \pm 0.2$ | $69.3 \pm 0.2$ | $76.3 \pm 0.1$ | $65.9 \pm 0.2$ |
| Fixed-label | $76.7 \pm 0.2$ | $66.5 \pm 0.1$ | $74.3 \pm 0.2$ | $64.2 \pm 0.2$ |
| w/o Curriculum | $77.5 \pm 0.1$ | $67.8 \pm 0.3$ | $75.7 \pm 0.2$ | $65.1 \pm 0.1$ |
| **Full model (Ours)** | $\mathbf{82.8 \pm 0.2}$ | $\mathbf{73.9 \pm 0.1}$ | $\mathbf{78.5 \pm 0.2}$ | $\mathbf{68.4 \pm 0.2}$ |

## 3.3 EMPIRICAL ANALYSIS

**a) Alignment of features.**: We plot t-SNE (Van der Maaten & Hinton, 2008) embeddings of features from the shared feature extractor (*i.e.,* the output of $\mathcal{F}$) across different source–target domain pairs from multiple datasets, as shown in Figure 3a. The features are extracted by passing adversarial examples generated using PGD-20 with $\epsilon = \frac{2}{255}$ through trained model. The visualizations clearly show that our method effectively aligns features across domains for both clean and adversarial samples.

**b) Analyzing Target Consistency for Convergence.** As shown in Figure 3b, the Target Consistency Rate increases steadily throughout training, indicating improved alignment of clean target samples with the multi-source latent space. This rising TCR reflects reduced classifier discrepancy and correlates with improved clean target accuracy. Furthermore, we continue to track TCR during implicit alignment, where a strong

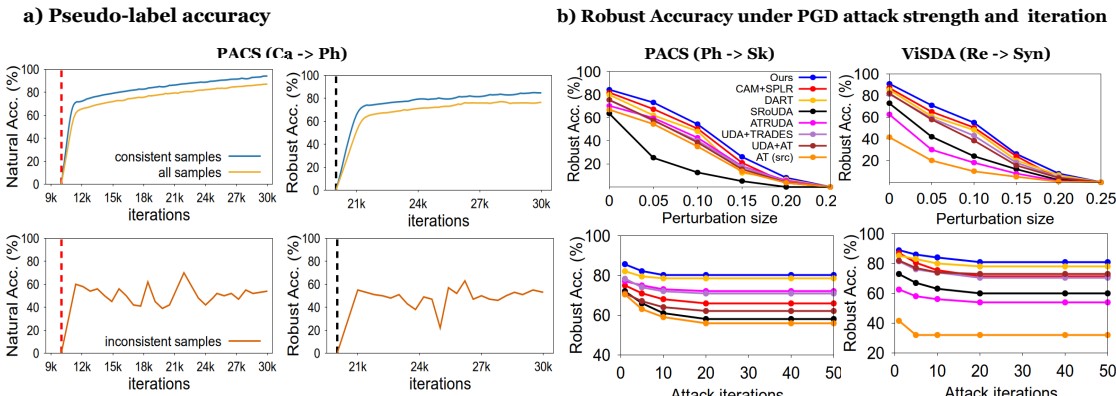

Figure 4: a) **Pseudo-label accuracy.** We observe higher pseudo-label accuracy for clean and adversarial target samples selected based on classifier consistency (top, blue curve), compared to those where classifier are not consistent (bottom). b) **Robust accuracy under different PGD attack strengths and iteration.** Our method consistently maintains higher robust accuracy as perturbation size (top) or attack iterations (bottom) increase.

correspondence between the consistency rate and adversarial target accuracy is observed due to consistent pseudo-labels support alignment between clean and adversarial features. These results validate that TCR serves as a reliable, label-free signal to determine convergence in both alignment phases.

**c) Pseudo-Label Accuracy under Target Consistency**: We use classifier consistency to select clean target samples for which both classifiers consistent over pseudo labels, forming the subset $\mathcal{D}_t^{(0)}$. In Figure 4, we evaluate pseudo-label accuracy for three groups: (i) the consistent subset $\mathcal{D}_t^{(0)}$, (ii) inconsistent samples where classifiers disagree, and (iii) the entire clean target set. The consistent subset exhibits noticeably higher pseudo-label accuracy, confirming that classifiers consistency is an effective indicator of pseudo-label reliability. In addition, we assess the pseudo-label accuracy of adversarial counterparts of $\mathcal{D}_t^{(0)}$ during the implicit alignment phase. These adversarial samples are introduced progressively using our curriculum strategy. We observe that pseudo-labels remain reliable when classifier consistency is maintained under perturbations (Figure 4a right), demonstrating the robustness of selection strategy.

**d) Impact of PGD attack budget and attack iteration.** We assess the robustness of our method on two source-target domain pairs from the VisDA and PACS datasets, comparing it against existing adversarially robust methods under varying PGD attack budgets. Specifically, we evaluate robustness by (i) increasing the perturbation size (Figure 4b top) and (ii) increasing the number of PGD steps while keeping $\epsilon = \frac{2}{255}$ fixed (Figure 4 b) bottom). As expected, robust accuracy declines with higher perturbation budgets and attack iterations, but our method consistently outperforms baselines across all settings. Analyses of curriculum learning, double consistency, and confidence-threshold ablations are provided in supplementary (Sec. E, E.2).

## 4 CONCLUSION

In this paper, we proposed an adversarially robust domain adaptation method from novel formulation of multi-source and multi-target adaptation. Our approach employs a progressive alignment strategy that combines explicit alignment with clean target and implicit alignment with adversarial target data under pseudo-label supervision. Our work can facilitate the development of effective and robust algorithms for unsupervised domain adaptation under adversarial settings. While our framework improves adversarial robustness, generating adversarial examples during training can increase computational cost and introduce optimization challenges. This trade-off is common in adversarial training methods and motivates future work on more efficient or adaptive adversarial training strategies.

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

**APPENDIX**

## A. RELATED WORK

### A.1 UNSUPERVISED DOMAIN ADAPTION

A central goal of UDA is to bridge the distributional gap between a labeled source domain and an unlabeled target domain by learning domain-invariant yet class-discriminative features. Inspired by the domain adaptation theory of Ben-David *et al.* Ben-David et al. (2010), numerous methods (Ganin et al., 2016b; Baktashmotlagh et al., 2016; Gong et al., 2012; Pan et al., 2010; Peng et al., 2019; Sun & Saenko, 2016b;a) have been proposed to align source and target distributions in the latent space. A popular direction leverages adversarial learning (Ganin et al., 2016a; Long et al., 2018a;b; Tzeng et al., 2017; Xu et al., 2018; Zhao et al., 2018), where a domain discriminator is trained to distinguish between domains while the feature extractor learns to deceive it. Class-conditional variants (Long et al., 2018a;b) enhance this framework by encouraging alignment conditioned on label predictions. Other alignment objectives focus on statistical discrepancy minimization between domains (Ganin et al., 2016b; Baktashmotlagh et al., 2016; Gong et al., 2012; Pan et al., 2010; Peng et al., 2019; Sun & Saenko, 2016b;a; Saito et al., 2018a), where alignment is achieved by minimizing measures such as moment differences or geodesic distances. For example, CORAL (Sun & Saenko, 2016a) reduces domain shift by aligning second-order statistics between domains, while MCD (Saito et al., 2018a) introduces a classifier discrepancy-based approach that exploits class decision boundary uncertainty to guide target alignment. An alternative line of work incorporates pseudo-labeling strategies (Xu et al., 2019; Choi et al., 2019; Yang et al., 2021a), where target labels are estimated using a source-trained model and refined iteratively. These methods often employ self-training or confidence-based filtering to mitigate noise in the pseudo-labels. While existing UDA methods focus on feature alignment under clean conditions, they often fail to ensure robustness against adversarial perturbations. This motivates the need for a unified framework that improves both adaptation performance and resilience against adversarial attacks.

### A.2 SUPERVISED ADVERSARIAL TRAINING

Adversarial Training (AT) (Madry et al., 2017) is a widely adopted defense mechanism against adversarial attacks. PGD-AT by Madry *et al.* Madry et al. (2017) formulates robustness as a max-min optimization problem, generating multi-step Projected Gradient Descent (PGD) adversarial examples and updating model

parameters using the most challenging perturbations. Although highly effective, PGD-AT often leads to a drop in clean accuracy. To mitigate this trade-off, TRADES (Zhang et al., 2020) introduces a theoretically grounded surrogate loss that explicitly separates natural and robust objectives, achieving a more balance between accuracy and robustness. MART (Wang et al., 2019) further improves robustness by focusing on misclassified examples, while FreeAT (Shafahi et al., 2019) reduces training cost by reusing gradient steps across mini-batches. More recent work, such as TWINS (Liu et al., 2023) enhances TRADES by promoting ensemble diversity, while recent methods such as (Li et al., 2024) incorporate self-guided label refinement, which refines soft label distributions without relying on external teachers and improves training through self-distillation. These approaches assume access to fully labeled training data, making them unsuitable for unsupervised domain adaptation (UDA), where target domain labels are not available. In contrast, our method is specifically designed for the UDA setting, aiming to improve robustness against adversarial attacks.

## A.3 ROBUST UDA METHODS

There has been extensive work on adversarial robustness in supervised learning, but relatively few methods explicitly address it in the context of Unsupervised Domain Adaptation (UDA) (Awais et al., 2021; Wang et al., 2024a; Lo & Patel, 2022; Zhu et al., 2023; Yang et al., 2021b; Soni & Dutta, 2025). RFA (Awais et al., 2021) aligns a UDA model to intermediate features from adversarially pretrained ImageNet teachers, improving robustness but assuming access to robust backbones, which is often infeasible in practical or low-resource domains. ARTUDA (Lo & Patel, 2022) generates adversarial examples from self-supervised signals and minimizes clean–adversarial logit discrepancy, but its reliance on pretext objectives limits generalization across diverse targets. Importantly, neither RFA nor ARTUDA explicitly treat clean and adversarial data as separate distributions to be aligned across domains. SRoUDA (Zhu et al., 2023) adopts a meta-learning strategy, assigning pseudo-labels with a pretrained UDA model before adversarial training, while DART (Wang et al., 2024a) generates adversarial target samples from pseudo-labels and retrains the model with combined classification and domain objectives. CAM+SPLR (Soni & Dutta, 2025) improves attention alignment alongside pseudo-label refinement, but like the others, remains sensitive to noisy pseudo-labels and treats robustness as an auxiliary task decoupled from adaptation. To the best of our knowledge, no prior work explicitly formulates robust UDA as a *multi-source, multi-target alignment problem* over clean source, adversarial source, clean target and adversarial target. This gap leaves cross-domain clean–adversarial distributions misaligned in the representation space and ultimately limits robust generalization.

## A.3 MULTI-SOURCE MULTI-TARGET DOMAIN ADAPTATION

Recent advances in multi-source multi-target domain adaptation (MSMTDA) (Lu et al., 2025; Zhu et al., 2019; Lu et al., 2024) tackle the challenge of aligning multiple source–target pairs using either domain-specific feature extractors or classifiers. Deng *et al.* Deng et al. (2023) propose a minimax framework that jointly estimates source–target mixture weights and predicts target models with convergence guarantees, while He *et al.* He et al. (2021) leverage ensemble pseudo-labels to constrain multiple adaptation models for better generalization. Other MSMTDA approaches (Wang et al., 2024b; Huang et al., 2025; Chen & Xiao, 2023; Tasar et al., 2020; Zheng et al., 2023; Wei, 2025) employ domain-specific classifiers or adaptation modules to reduce discrepancies across domains. However, such pairwise alignment strategies often create fragmented decision boundaries and do not account for adversarial robustness. In contrast, we adopt a common feature extractor with shared classifiers to stabilize the latent space across multiple sources and progressively align clean and adversarial domains across targets, providing the first MSMTDA-based framework that explicitly integrates robustness into adaptation.

# B. ARCHITECTURE AND IMPLEMENTATION DETAILS

**Datasets.** We present the results of our approach on four standard benchmark datasets*OfficeHome* (Wang et al., 2021) which has four domains across 65 categories *i.e.,* Art (Ar, 2427 images), ClipArt (Cl, 4365 images), Product(Pr, 4439 images) and RealWorld (Re, 4357 images), *PACS* (Li et al., 2017) has four domains with seven categories, namely Photo (Ph, 1670 images), Art Painting (Ar, 2048 images), Cartoon (Ca, 2344 images) and Sketch (Sk, 3929 images), and *VisDA* (Peng et al., 2017) is a large dataset having two domains with 12 categories namely, Synthetic images (Syn, 152409) and Real images (Re, 55400). ) *Digit* dataset containing three different domains across 10 classes, MNIST (M, 64015 images), USPS (U, 9078 images), and SVHN (S, 96322 images). Note that the images in M, U are gray-scale, whereas the images in S are colored.

**Prior Works.** We compare our method against a range of standard UDA baselines, source-only adversarial training, target-only adversarial training, and recent state-of-the-art robust domain adaptation approaches. The methods include: Maximum Mean Discrepancy-based Domain Adaptation (MDD) (Zhang et al., 2019b), Smooth Domain Adversarial Training (SDAT) (Rangwani et al., 2022), Low-Uncertainty High-Potential Instance Learning (LUHP) (Zhang et al., 2024),Domain-Adversarial Neural Network (DANN) (Ganin et al., 2016a), Adversarial Training (AT(src only)) (Madry et al., 2017), UDA with standard adversarial training (UDA+AT) (Madry et al., 2017), UDA with TRadeoff-inspired Adversarial Defense via Surrogate-loss minimization (UDA+TRADES) (Zhang et al., 2019a), Adversarially Robust Training for Unsupervised Domain Adaptation (ARTUDA) (Yang et al., 2021b), Meta Self-training for Robust Unsupervised Domain Adaptation (SRoUDA) (Zhu et al., 2023), Divergence-Aware Adversarial Training (DART) (Wang et al., 2024a), and Consistent Attention Mapping with Self Pseudo-Label Refinement (CAM+SPLR) (Soni & Dutta, 2025). DANN are standard UDA methods that focus on source–target alignment but do not address adversarial robustness. AT(src) applies adversarial training only on labeled source data without using any target information. UDA+AT and UDA+TRADES extend DANN by first generating pseudo-labels for target domain and then applying adversarial training using cross-entropy and TRADES losses, respectively. ARTUDA, SRoUDA, DART, and CAM+SPLR are recent robust UDA methods explicitly designed to improve performance under adversarial domain shifts.

## B.1 MODEL ARCHITECTURE

Our model comprises two main components: a backbone feature extractor and two classifier heads. These components enable effective representation learning and facilitate the training of our proposed robust domain adaptation framework.

**Feature Extractor.** For all datasets except Digit, we use a ResNet-50 backbone pretrained on ImageNet (Deng et al., 2009). The final fully connected classification layer is removed to obtain a 2048-dimensional feature representation. Given a mini-batch of size $B$, the output is a globally average pooled feature map of shape $(B, 2048)$, serving as a high-level, domain invariant feature representation.

**Digit Datasets.** For Digit datasets, we adopt the convolutional neural network architecture that has been used in (Gulrajani & Lopez-Paz, 2020). This lightweight CNN (see in Table 3) consists of four convolutional blocks followed by global average pooling.

The resulting 128-dimensional feature vector is forwarded to two classifier heads for further processing.

**Two Classifier Heads.** We append two parallel classifier heads, $\mathcal{H}_1$ and $\mathcal{H}_2$, each implemented as a linear layer with an output dimension equal to the number of classes (specific to the dataset). Both heads operate on the 2048-dimensional feature representation obtained from the shared feature extractor, except for the Digit datasets, where the feature dimension is 128. Each classifier head is implemented using a single fully

Table 3: Convolutional neural network architecture used for DIGIT datasets (MNIST, SVHN, USPS). All convolution layers use $3 \times 3$ kernels with "same" padding.

| Layer | Description |
|-------|-------------|
| 1 | Conv2D (in: $d$, out: 64), kernel: $3 \times 3$, stride: 1, padding: same |
| 2 | ReLU |
| 3 | GroupNorm (8 groups) |
| 4 | Conv2D (in: 64, out: 128), kernel: $3 \times 3$, stride: 2, padding: same |
| 5 | ReLU |
| 6 | GroupNorm (8 groups) |
| 7 | Conv2D (in: 128, out: 128), kernel: $3 \times 3$, stride: 1, padding: same |
| 8 | ReLU |
| 9 | GroupNorm (8 groups) |
| 10 | Conv2D (in: 128, out: 128), kernel: $3 \times 3$, stride: 1, padding: same |
| 11 | ReLU |
| 12 | GroupNorm (8 groups) |
| 13 | Global average pooling |

connected layer : `nn.Linear(2048, K)` for non digit datasets and `nn.Linear(128, K)` for Digit datasets, where K denotes the number of classes.

## B.2 TRAINING DETAILS AND HYPERPARAMETERS

**Data Preprocessing and Augmentation.** We follow the standard data augmentation pipeline introduced by (Gulrajani & Lopez-Paz, 2020) for all datasets except Digit benchmarks. The same augmentations are applied to both labeled source data and unlabeled target data. Specifically, each image undergoes a random resized crop (with varying size and aspect ratio), followed by resizing to $224 \times 224$ pixels. Additional augmentations include random horizontal flipping, color jittering, and random grayscaling with a 10% probability. All images are then normalized using the ImageNet channel-wise mean and standard deviation.

For Digit datasets, we apply minimal preprocessing, which includes resizing all images to $32 \times 32$ and converting them to 3-channel grayscale images, followed by normalization to a fixed mean and standard deviation of 0.5 across channels.

**Data Splits.** For the source domain, we retain 80% of the data for training. During the warm-start, a small 10% subset of the source training data is further reserved as a validation set to monitor early training and select initial model checkpoints before the adaptation stage begins. For the target domain, we split the data into an unlabeled training set and a held-out test set using an 8:2 ratio. The unlabeled training set is used for adaptation, while the test set is used exclusively for final evaluation.

**Weight Initialization.** The ResNet-50 backbone is initialized with ImageNet-pretrained weights. The two classifier layers are initialized independently using Xavier uniform initialization for the weights, and all biases are initialized to zero. This promotes diversity across classifiers, enabling effective discrepancy-based alignment and robust pseudo-labeling.

**Adversarial Attack Setting.** We use an $\ell_\infty$-norm perturbation set defined as $\mathcal{B}(\mathbf{x}) = \{\hat{\mathbf{x}} : \|\hat{\mathbf{x}} - \mathbf{x}\|_\infty \leq \epsilon\}$ in our setting. During training, adversarial examples are generated using 5-step PGD with a step size of $\alpha = \frac{1}{255}$ and a maximum perturbation of $\epsilon = \frac{2}{255}$. This adversarial training setup helps the model learn robust representations by exposing it to worst-case perturbations during optimization. For evaluation, we report both natural accuracy and robust accuracy on the target data. Robust accuracy is computed using PGD-20 attacks with perturbation sizes $\epsilon = \frac{2}{255}$ and $\epsilon = \frac{8}{255}$, to assess performance under varying threat levels.

**Hyperparameters.** Tables 4 and 5 summarize the hyperparameters used in our training pipeline for the non-digit and digit datasets, respectively. See Code [1]

Table 4: Hyperparameters and its value used in experiments for Non-Digit Datasets.

| Hyperparameter | Description | Value |
|---|---|---|
| $n$ | Epoch interval to recompute pseudo-labels | 10 |
| batch_size | Batch size for both source and target | 16 |
| learning_rate | Learning rate for Adam optimizer | $1 \times 10^{-4}$ |
| weight_decay | Weight decay regularization | $5 \times 10^{-5}$ |
| Adam $\beta_1$ | First moment decay in Adam optimizer | 0.9 |
| stop_TCR | TCR threshold to stop alignment | $\geq 0.99$ |
| stagnate_count | Patience for stagnation in training | 3 epochs |

Table 5: Hyperparameters and its value used in experiments for Digit Dataset.

| Hyperparameter | Description | Value |
|---|---|---|
| $n$ | Epoch interval to recompute pseudo-labels | 10 |
| batch_size | Batch size for both source and target | 64 |
| learning_rate | Learning rate for Adam optimizer | $1 \times 10^{-3}$ |
| weight_decay | Weight decay regularization | 0 |
| Adam $\beta_1$ | First moment decay in Adam optimizer | 0.9 |
| stop_TCR | TCR threshold to stop alignment | $\geq 0.99$ |
| stagnate_count | Patience for stagnation in training | 3 epochs |

## C. ALGORITHM

The full training pipeline is provided in Algorithm 1.

## D. ABLATION RESULTS UNDER PGD

This section presents results on target test data under varying PGD-20 attack strengths ($\epsilon = \frac{2}{255}, \frac{8}{255}$), as well as on the source test data. [†] Results are not given in mean ± standard deviation format.

**Note:** All equation references (e.g., Eq. (3), Eq. (5)) refer to those defined in the main paper.

### D.1 RESULTS UNDER PGD-20 ATTACK WITH $\epsilon = \frac{2}{255}$

Owing to space constraints, the main paper presents the results averaged across all source–target domain pairs for the Office-Home, PACS, VisDA, and Digit datasets. In this supplementary material, we provide the complete results for all individual source–target pairs. Specifically, We report the Natural accuracy and robust accuracy on all source-target domain pairs under the PGD-20 attack with $\epsilon = \frac{2}{255}$ across each dataset. Across the 29 evaluated source–target domain pairs, our method achieves the highest robust target accuracy in 26 pairs under PGD-20 attack with $\epsilon = \frac{2}{255}$ and outperforms all baselines on clean target accuracy in 24 pairs.

---

[1]Our source code is available at: https://github.com/Researcharhieve9/adversarial-vision

---

**Algorithm 1:** Progressive Alignment for Robust Domain Adaptation

---

**Input:** Source datasets $\mathcal{D}_s^{\text{cln}}, \mathcal{D}_s^{\text{adv}}$; Unlabeled target dataset $\mathcal{D}_t^{cln}, \mathcal{D}_t^{\text{adv}}$; Model $(\mathcal{F}, \mathcal{H}_1, \mathcal{H}_2)$
**Output:** Robust Adapted model $(\mathcal{F}, \mathcal{H}_1, \mathcal{H}_2)$

**1 for** *epoch* $e = 1$ *to* $E_{warm}$ **do**
**2**     Load mini-batches $B_s^{\text{cln}}$ from $\mathcal{D}_s^{\text{cln}}$ ;
**3**     Generate adversarial batch $B_s^{\text{adv}}$ using PGD$(B_s^{\text{cln}}; \mathcal{F}, \mathcal{H}_1, \mathcal{H}_2)$ ;         `// Warm-start`
**4**     Update $(\mathcal{F}, \mathcal{H}_1, \mathcal{H}_2)$ using source objective (Eq. (3)) ;

**5 while** $\mathcal{C}(\mathcal{D}_t^{cln})$ *has not converged* **do**
**6**     Load mini-batches $B_s^{\text{cln}}$ from $\mathcal{D}_t^{\text{cln}}$ ;         `// Explicit alignment`
**7**     Update classifiers $\mathcal{H}_1, \mathcal{H}_2$ to maximize discrepancy (Eq. (4)) ;
**8**     Update feature extractor $\mathcal{F}$ to minimize discrepancy (Eq. (5)) ;

**9** Obtain pseudo-labeled subset $\mathcal{D}_t^{(0)}$ using classifier consistency on predicted label. (Eq. (9)) ;
**10** Prepare reliable subset $\mathcal{D}_t^{\text{adv}}$ using double consistency on adversarial examples (Eq. (10)) ;
**11** Form curriculum buffer $\mathcal{D}_t^{\text{curr}}$ by sorting $\mathcal{D}_t^{(0)} \setminus \mathcal{D}_t^{\text{adv}}$ by descending confidence (Eq. (11)) ;
**12 while** $\mathcal{C}(\mathcal{D}_t^{cln})$ *has not converged* **do**
**13**     Load mini-batches $B_s^{\text{cln}}$ from $\mathcal{D}_s^{\text{cln}}$ ;
**14**     Generate adversarial batch $B_s^{\text{adv}}$ using PGD$(B_s^{\text{cln}}; \mathcal{F}, \mathcal{H}_1, \mathcal{H}_2)$ ;     `// Implicit alignment`
**15**     Update $(\mathcal{F}, \mathcal{H}_1, \mathcal{H}_2)$ using source objective (Eq. (3)) ;
**16**     Load mini-batches $B_t^{\text{cln}}$ from $\mathcal{D}_t^{\text{adv}} \cup \mathcal{D}_t^{\text{curr}}$ ;
**17**     Generate adversarial target batch $B_t^{\text{adv}}$ using PGD$(B_t^{\text{cln}}; \mathcal{F}, \mathcal{H}_1, \mathcal{H}_2)$ ;
**18**     Update $(\mathcal{F}, \mathcal{H}_1, \mathcal{H}_2)$ using target objective (Eq. (12)) ;
**19**     **if** $n$ *epochs have passed since last update* **then**
**20**        Recompute $\mathcal{D}_t^{(0)}, \mathcal{D}_t^{\text{adv}}$, and $\mathcal{D}_t^{\text{curr}}$ (Lines 9 to 11) ;

---

## 1. DIGIT

Table 6 presents the results for three source–target domain pairs. n this dataset, our method demonstrates competitive performance on both natural and robust accuracy compared to existing methods. The domain gap between MNIST and USPS is relatively small, allowing our method to match the performance of prior approaches. In contrast, the MNIST to SVHN pair exhibits a larger domain gap, where our method achieves slight improvements in robustness over DART (Wang et al., 2024a).

Table 6: Natural accuracy (Nat.) and robust accuracy (PGD) on the target test data of the DIGIT dataset, evaluated on three source-target pairs under a PGD-20 attack with $\epsilon = \frac{2}{255}$.

| S $\rightarrow$ T | M$\rightarrow$U | | U$\rightarrow$M | | M$\rightarrow$S | |
|---|---|---|---|---|---|---|
| **Method** | Nat. | PGD | Nat. | PGD | Nat. | PGD |
| DANN | 98.8±0.2 | 97.2±0.4 | 98.3±0.1 | 95.9±0.4 | 21.8±0.2 | 12.4±0.6 |
| AT (src only) | 99.0±0.1 | 98.1±0.1 | 98.3±0.0 | 97.5±0.0 | **25.5 ±1.2** | 14.4±0.1 |
| UDA + AT | 99.0±0.2 | 98.4±0.2 | 98.5±0.1 | 98.2±0.1 | 22.9±0.0 | 17.9±0.1 |
| UDA + TRADES | 99.1±0.2 | 98.6±0.2 | 98.5±0.1 | 98.2±0.1 | 22.5±0.1 | 17.9±0.2 |
| ARTUDA | **99.2±0.1** | **98.8±0.2** | 99.0±0.0 | **98.8±0.0** | 20.0±0.4 | 18.7±0.1 |
| SROUDA | 99.0±0.2 | 98.5±0.2 | 98.4±0.0 | 97.9±0.1 | 23.0±1.8 | 18.8±0.4 |
| DART | 99.1±0.1 | 98.5±0.1 | 98.8±0.0 | 98.4±0.0 | 22.5±0.1 | 19.1±0.3 |
| **Ours** | 99.0 ± 0.1 | 98.6 ± 0.1 | **99.1 ± 0.1** | 98.6 ± 0.1 | 23.1 ± 0.1 | **19.1 ± 0.2** |

## 2. PACS

Table 7, 8, 9, 10 presents the results for 12 source-target domain pairs. Our method outperforms existing approaches in robustness across all source–target domain pairs, owing to its effective alignment of both clean and adversarial target features.

Table 7: Natural accuracy (Nat.) and Robust accuracy (PGD) on the target test data of the PACS dataset with Photo as the source domain, evaluated under a PGD-20 attack with $\epsilon = \frac{2}{255}$.

| S → T | Ph → Ar | | Ph → Ca | | Ph → Sk | |
|---|---|---|---|---|---|---|
| Method | Nat. | PGD | Nat. | PGD | Nat. | PGD |
| DANN | 89.1±0.2 | 3.9±3.1 | 80.5±0.2 | 11.5±2.5 | 74.0±1.1 | 24.0±2.1 |
| AT(src only) | 71.0±3.0 | 32.7±1.6 | 71.4±1.9 | 50.4±0.6 | 69.3±0.5 | 61.0±0.6 |
| UDA+AT | 82.3±0.7 | 59.1±0.7 | 85.5±0.6 | 77.4±1.0 | 78.2±0.4 | 75.5±0.3 |
| UDA + TRADES | 82.1±1.0 | 63.2±1.5 | 84.4±0.2 | 76.7±0.9 | 78.7±0.5 | 75.3±0.7 |
| ARTUDA | 85.9±1.1 | 60.1±1.4 | 87.5±1.7 | 78.1±0.5 | 74.9±1.3 | 70.4±1.4 |
| SRoUDA | 76.1±1.7 | 56.4±0.2 | 82.4±1.3 | 71.7±1.8 | 71.9±0.9 | 63.7±1.2 |
| DART | 85.2±1.2 | 58.0±0.9 | 89.4±0.8 | 80.5±0.3 | 82.5±0.8 | 79.9±0.4 |
| CAM+SPLR[†] | 87.6 | 63.4 | **92.5** | 82.6 | 83.2 | 81.9 |
| **Ours** | **89.3±0.2** | **66.1 ± 0.3** | 91.2±0.2 | **84.6 ± 0.3** | **84.2 ± 0.3** | **82.4 ± 0.2** |

Table 8: Natural and PGD accuracy of target test data on the PACS dataset with Cartoon as the source domain, evaluated under a PGD-20 attack with $\epsilon = \frac{2}{255}$.

| S → T | Ca → Ar | | Ca → Ph | | Ca →Sk | |
|---|---|---|---|---|---|---|
| Method | Nat. | PGD | Nat. | PGD | Nat. | PGD |
| DANN | 84.9±0.7 | 0.6±0.3 | 92.5±0.7 | 1.4±0.4 | 78.2±0.9 | 25.7±2.2 |
| AT(src only) | 59.6±1.0 | 30.2±1.0 | 77.9±1.9 | 53.8±0.5 | 77.1±0.9 | 66.6±0.6 |
| UDA+AT | 76.2±1.7 | 55.0±1.6 | 93.3±0.5 | 80.3±0.8 | 80.0±0.3 | 77.4±0.2 |
| UDA + TRADES | 78.5±1.7 | 58.0±1.4 | 92.2±0.1 | 82.1±0.5 | 79.9±0.5 | 77.6±0.4 |
| ARTUDA | 76.5±2.5 | 53.3±1.6 | 89.4±0.9 | 75.0±1.7 | 80.3±0.4 | 74.9±1.0 |
| SRoUDA | 72.0±1.5 | 50.9±1.1 | 90.3±0.9 | 79.9±2.0 | 76.7±1.2 | 72.3±1.3 |
| DART | 77.4±1.1 | 54.6±0.3 | 94.2±0.5 | 79.8±1.2 | 84.9±0.4 | 81.0±0.7 |
| CAM+SPLR[†] | 81.8 | 61.5 | **94.9** | 83.5 | 85.3 | 82.6 |
| **Ours** | **84.7 ± 0.2** | **64.3 ± 0.1** | 93.8 ± 0.2 | **85.0 ± 0.2** | **86.2 ± 0.3** | **83.4 ± 0.3** |

Table 9: Natural accuracy (Nat.) and Robust accuracy (PGD) on the target test data of the PACS dataset with Art as the source domain, evaluated under a PGD-20 attack with $\epsilon = \frac{2}{255}$.

| S → T | Ar → Ca | | Ar → Ph | | Ar → Sk | |
|---|---|---|---|---|---|---|
| Method | Nat. | PGD | Nat. | PGD | Nat. | PGD |
| DANN | 84.3±0.6 | 12.4±6.1 | 97.9±0.4 | 2.7±1.5 | 84.9±0.7 | 0.6±0.3 |
| AT(src only) | 79.4±0.4 | 63.9±1.2 | 82.5±0.6 | 65.8±0.5 | 79.9±1.1 | 72.4±1.2 |
| UDA+AT | 85.1±0.1 | 76.5±0.9 | 95.0±1.4 | 83.1±1.2 | 84.8±0.1 | 81.1±0.6 |
| UDA+TRADES | 85.1±0.9 | 77.2±1.0 | 95.3±0.7 | 82.2±0.6 | 86.1±0.7 | 83.3±0.7 |
| ARTUDA | 88.3±2.1 | 76.0±1.7 | 95.0±0.6 | 78.5±1.3 | 80.3±1.3 | 61.5±1.0 |
| SRoUDA | 84.2±1.4 | 75.8±0.6 | 94.1±0.7 | 81.5±1.0 | 77.3±4.6 | 73.2±4.9 |
| DART | 89.1±0.3 | 79.1±0.3 | 95.9±0.7 | 81.4±1.4 | 89.5±0.6 | 86.4±0.5 |
| CAM+SPLR[†] | 90.5 | 83.1 | 97.5 | 85.0 | 90.8 | 86.7 |
| **Ours** | **91.6 ± 0.3** | **84.9 ± 0.3** | **98.1 ± 0.1** | **85.8 ± 0.3** | **92.3 ± 0.2** | **88.4 ± 0.3** |

Table 10: Natural accuracy (Nat.) and robust accuracy (PGD) on the target test data of the PACS dataset with Photo (Ph) as the source domain, evaluated under a PGD-20 attack with $\epsilon = \frac{2}{255}$.

| S → T | Sk → Ar | | Sk → Ca | | Sk → Ph | |
|---|---|---|---|---|---|---|
| Method | Nat. | PGD | Nat. | PGD | Nat. | PGD |
| DANN | 68.0±1.2 | 1.5±1.2 | 72.3±1.4 | 14.0±4.5 | 71.1±4.0 | 0.3±0.1 |
| AT(src only) | 22.3±1.4 | 19.0±0.9 | 66.6±1.7 | 40.2±1.1 | 31.7±5.5 | 22.7±1.1 |
| UDA+AT | 62.4±2.2 | 37.5±0.8 | 72.5±1.8 | 64.0±1.2 | 88.8±0.7 | 73.6±0.7 |
| UDA+TRADES | 69.1±2.3 | 44.2±2.5 | 71.6±0.6 | 63.8±0.6 | 89.6±0.7 | 76.4±1.4 |
| ARTUDA | 49.5±2.4 | 31.7±3.4 | 38.1±2.5 | 25.5±1.8 | 48.9±1.8 | 40.4±2.9 |
| SRoUDA | 24.5±1.7 | 22.4±0.3 | 72.4±1.0 | 62.3±0.2 | 91.9±0.5 | 73.1±3.6 |
| DART | 71.9±1.8 | 53.1±4.4 | 78.4±0.7 | 69.2±0.9 | 87.8±1.4 | 76.8±1.0 |
| CAM+SPLR[†] | 73.5 | 56.9 | 81.6 | 73.8 | 89.4 | 79.2 |
| **Ours** | **75.3 ± 0.1** | **59.4 ± 0.3** | **82.8 ± 0.2** | **73.9 ± 0.1** | **91.8 ± 0.3** | **81.0 ± 0.2** |

## 3. OFFICE-HOME

Table 11, 12, 13, 14 presents the results for 12 source-target domain pairs. Our method demonstrates strong performance in both natural and robust accuracy. While CAM+SPLR provides competitive results in terms of robust accuracy on the Product → Clipart pair and natural accuracy on the RealWorld → Clipart pair, our approach consistently achieves better overall performance. However, standard domain-shift UDA methods such as MDD, SDAT, and LUHP achieve strong performance in clean settings due to their effective feature-alignment mechanisms. In contrast, our method simultaneously maintains high clean accuracy while substantially improving robustness under adversarial perturbations, demonstrating its advantage in scenarios where both domain shift and adversarial attacks must be tackled together.

Table 11: Natural accuracy (Nat.) and robust accuracy (PGD) on the target test data of the Office-home dataset with Product (Pr) as the source domain, evaluated under a PGD-20 attack with $\epsilon = \frac{2}{255}$.

| S → T | Pr → Ar | | Pr → Cl | | Pr → Re | |
|---|---|---|---|---|---|---|
| Method | Nat. | PGD | Nat. | PGD | Nat. | PGD |
| MDD | 61.2±0.0 | – | 53.6±0.0 | – | 60.4±0.0 | – |
| SDAT | 63.3±0.0 | – | 57.0±0.0 | – | 82.2±0.0 | – |
| LUHP | **70.5±0.0** | – | **60.7±0.0** | – | **84.±0.0** | – |
| DANN | 49.1±0.3 | 0.2±0.1 | 57.4±0.2 | 2.1±0.5 | 60.0±0.6 | 0.3±0.1 |
| AT(src only) | 33.8±1.3 | 15.3±0.4 | 47.2±0.1 | 34.1±0.6 | 56.9±1.3 | 34.6±1.0 |
| UDA+AT | 38.5±1.6 | 18.3±0.6 | 49.1±0.8 | 42.9±0.4 | 61.4±1.5 | 44.2±1.2 |
| UDA+TRADES | 37.9±2.2 | 20.5±0.8 | 49.3±1.1 | 44.3±1.7 | 61.6±0.9 | 44.0±1.5 |
| ARTUDA | 38.3±2.1 | 18.0±1.4 | 48.5±0.9 | 42.8±0.8 | 62.4±0.3 | 42.7±2.0 |
| SRoUDA | 33.5±1.3 | 22.4±1.3 | 49.9±0.4 | 41.6±0.6 | 60.2±2.0 | 45.6±0.7 |
| DART | 43.7±2.5 | 21.5±0.8 | 52.5±1.3 | 44.8±1.3 | 63.5±0.8 | 43.6±0.5 |
| CAM+SPLR[†] | 45.9 | 28.8 | 54.5 | **48.0** | 64.6 | 49.1 |
| **Ours** | 50.4 ± 0.2 | **32.3±0.1** | 58.0±0.2 | 46.6 ±0.2 | 70.1±0.3 | **54.7±0.2** |

## 4. VISDA

Table 15 represents the results for 2 source-target domain pairs. Our method outperforms existing approaches in both natural and robust accuracy across all source–target domain pairs. While prior approaches achieve competitive results on Real → Synthetic domain pairs, they fail to generalize as effectively across other domain pair.

Table 12: Natural accuracy (Nat.) and robust accuracy (PGD) on the target test data of the Office-home dataset with Clipart (Cl) as the source domain, evaluated under a PGD-20 attack with $\epsilon = \frac{2}{255}$.

| S → T | Cl → Ar | | Cl → Pr | | Cl → Re | |
|---|---|---|---|---|---|---|
| **Method** | Nat. | PGD | Nat. | PGD | Nat. | PGD |
| MDD | 60.0±0.0 | – | 71.4±0.0 | – | 71.8±0.0 | – |
| SDAT | 66.3±0.0 | – | 77.6±0.0 | – | 76.8±0.0 | – |
| LUHP | **72.5±0.0** | – | **81.7±0.0** | – | **81.0±0.0** | – |
| DANN | 45.2±0.8 | 0.0±0.0 | 47.9±0.8 | 3.6±1.0 | 67.4±1.5 | 0.6±0.3 |
| AT(src only) | 34.4±1.8 | 14.5±0.6 | 51.2±1.5 | 33.1±0.8 | 53.8±1.0 | 28.3±0.1 |
| UDA+AT | 39.4±1.5 | 23.0±0.1 | 55.6±0.8 | 46.8±0.2 | 56.5±0.8 | 41.5±0.4 |
| UDA+TRADES | 40.0±1.0 | 22.0±0.4 | 56.2±0.3 | 47.9±0.2 | 56.2±0.3 | 43.8±1.0 |
| ARTUDA | 42.0±0.2 | 20.2±1.0 | 56.1±1.3 | 44.1±1.4 | 58.9±1.2 | 30.2±0.6 |
| SRoUDA | 36.3±0.3 | 23.8±0.6 | 53.9±1.0 | 47.2±1.4 | 55.1±1.7 | 42.1±0.8 |
| DART | 44.1±0.9 | 24.2±0.5 | 57.0±0.3 | 45.5±0.6 | 57.8±0.3 | 39.6±0.2 |
| CAM+SPLR[†] | 45.4 | 28.0 | 58.5 | 49.0 | 62.5 | 48.0 |
| **Ours** | 50.7 ± 0.3 | **35.0 ± 0.2** | 62.7 ± 0.2 | **52.5 ± 0.3** | 63.1 ± 0.2 | **49.3 ± 0.3** |

Table 13: Natural accuracy (Nat.) and robust accuracy (PGD) on the target test data of the Office-home dataset with Realworld (Re) as the source domain, evaluated under a PGD-20 attack with $\epsilon = \frac{2}{255}$.

| S → T | Re → Ar | | Re → Cl | | Re → Pr | |
|---|---|---|---|---|---|---|
| **Method** | Nat. | PGD | Nat. | PGD | Nat. | PGD |
| MDD | 72.5±0.0 | – | 60.2±0.0 | – | 82.3±0.0 | – |
| SDAT | 74.9±0.0 | – | 64.7±0.0 | – | 86.0±0.0 | – |
| LUHP | **75.9±0.0** | – | **65.2±0.0** | – | **85.6±0.0** | – |
| DANN | 66.0±0.6 | 0.4±0.1 | 55.5±0.6 | 3.6±0.5 | 74.3±0.9 | 1.8±0.2 |
| AT(src only) | 47.2±1.2 | 28.0±1.2 | 54.1±0.8 | 40.9±1.0 | 66.7±0.3 | 49.9±1.1 |
| UDA+AT | 46.4±0.8 | 29.4±1.4 | 55.0±0.5 | 49.4±0.6 | 72.3±1.5 | 60.4±0.8 |
| UDA+TRADES | 47.9±2.4 | 27.4±0.4 | 55.6±0.9 | 49.7±0.8 | 70.0±1.4 | 61.9±1.2 |
| ARTUDA | 49.5±2.0 | 28.4±1.0 | 58.3±0.7 | 48.5±0.9 | 73.0±0.6 | 58.3±0.4 |
| SRoUDA | 42.1±1.4 | 27.5±0.1 | 55.4±0.6 | 47.3±0.7 | 70.7±0.6 | 60.6±1.2 |
| DART | 53.7±1.0 | 29.1±1.6 | 58.6±0.4 | 49.8±1.1 | 74.0±0.9 | 60.2±1.5 |
| CAM+SPLR[†] | 58.2 | 35.5 | 62.4 | 53.4 | 73.6 | 64.5 |
| **Ours** | 63.2 ± 0.2 | **42.5 ± 0.1** | 59.5 ± 0.1 | **54.3 ± 0.3** | 78.5 ± 0.2 | **68.4 ± 0.3** |

## D.2 RESULTS UNDER PGD-20 ATTACK WITH $\epsilon = \frac{8}{255}$

The results for three source–target domain pairs and two domain pairs under the PGD-20 attack with $\epsilon = \frac{8}{255}$ are reported for the *PACS* (Table 17) and *Digit* (Table 18) datasets, respectively. These results demonstrate that our method achieves strong performance in both robust and natural accuracy under perturbation strength ($\epsilon = \frac{8}{255}$).

## D.3 RESULTS ON SOURCE TEST DATA

In Table 19, we report Natural and robust accuracy under the PGD-20 attack with $\epsilon = \frac{8}{255}$ on the source test data. These results clearly demonstrate that our method, when applied to the source domain, consistently maintains or even improves robustness on the source test set. AT (src only) provides competitive robustness on the source test data, while DART (Wang et al., 2019) also improved the robust accuracy compared to other existing methods.

Table 14: Natural accuracy (Nat.) and robust accuracy (PGD) of target test data for source domain Art (Ar) on the Office-Home dataset, evaluated under a PGD-20 attack with $\epsilon = \frac{2}{255}$.

| S → T | Ar → Cl | | Ar → Pr | | Ar → Re | |
|---|---|---|---|---|---|---|
| **Method** | Nat. | PGD | Nat. | PGD | Nat. | PGD |
| MDD | 54.9±0.0 | – | 73.7±0.0 | – | 77.8±0.0 | – |
| SDAT | 58.2±0.0 | – | 77.1±0.0 | – | 82.2±0.0 | – |
| LUHP | **72.5±0.0** | – | **81.7±0.0** | – | **81.0±0.0** | – |
| DANN | 49.1±0.3 | 2.6±0.4 | 55.5±0.8 | 0.9±0.2 | **66.8**±0.8 | 1.0±0.3 |
| AT(src only) | 45.4±0.6 | 32.0±0.3 | 48.5±0.4 | 29.8±0.3 | 57.2±2.1 | 36.1±0.8 |
| UDA+AT | 48.0±0.5 | 41.7±0.7 | 55.9±0.4 | 46.2±0.3 | 57.6±0.6 | 40.5±1.1 |
| UDA+TRADES | 48.6±0.3 | 43.6±0.4 | 55.9±0.4 | 47.6±0.1 | 57.1±1.6 | 41.8±0.2 |
| ARTUDA | 50.9±1.6 | 41.7±1.7 | 55.0±0.8 | 41.2±1.0 | 61.7±0.6 | 42.5±1.0 |
| SRoUDA | 48.2±0.5 | 38.9±0.5 | 52.9±0.6 | 45.8±0.3 | 57.4±1.4 | 44.2±0.7 |
| DART | 50.4±0.9 | 42.2±0.6 | 60.1±0.2 | 47.7±1.0 | 62.7±0.5 | 40.7±0.5 |
| CAM+SPLR[†] | 52.8 | 45.3 | 61.5 | 52.9 | 64.6 | 47.9 |
| **Ours** | $55.6 \pm 0.3$ | $\mathbf{47.8 \pm 0.2}$ | $64.7 \pm 0.3$ | $\mathbf{55.6 \pm 0.3}$ | $67.5 \pm 0.3$ | $\mathbf{53.4 \pm 0.2}$ |

Table 15: Natural accuracy (Nat.) and robust accuracy (PGD) on the VISDA dataset, evaluated under a PGD-20 attack with $\epsilon = \frac{2}{255}$.

| S → T | Syn → Re | | Re → Syn | |
|---|---|---|---|---|
| **Method** | Nat. | PGD | Nat. | PGD |
| DANN | $67.4 \pm 0.2$ | $0.5 \pm 0.2$ | $78.6 \pm 0.9$ | $0.8 \pm 0.1$ |
| AT (src only) | $19.0 \pm 0.2$ | $18.0 \pm 0.3$ | $53.5 \pm 0.8$ | $41.6 \pm 0.4$ |
| UDA+AT | $69.6 \pm 0.3$ | $58.3 \pm 0.7$ | $85.7 \pm 0.2$ | $82.0 \pm 0.2$ |
| UDA+TRADES | $68.1 \pm 0.7$ | $57.9 \pm 0.5$ | $85.1 \pm 0.3$ | $81.5 \pm 0.5$ |
| ARTUDA | $45.2 \pm 4.8$ | $32.5 \pm 2.7$ | $72.5 \pm 2.5$ | $62.6 \pm 0.3$ |
| SRoUDA | $48.2 \pm 2.7$ | $33.4 \pm 0.7$ | $81.2 \pm 1.4$ | $72.9 \pm 1.3$ |
| DART | $69.5 \pm 0.2$ | $58.0 \pm 0.5$ | $87.3 \pm 0.3$ | $85.3 \pm 0.2$ |
| CAM+SPLR[†] | 72.8 | 65.9 | 89.5 | 87.1 |
| **Ours** | $\mathbf{75.3 \pm 0.1}$ | $\mathbf{66.6 \pm 0.2}$ | $\mathbf{89.7 \pm 0.1}$ | $\mathbf{88.0 \pm 0.2}$ |

## E. MORE ANALYSIS

This section presents the analysis of the double consistency and curriculum strategies. Additionally, we perform an ablation study by removing the double consistency mechanism and replacing it with a confidence-based thresholding scheme.

### E.1 IMPACT OF DOUBLE CONSISTENCY AND CURRICULUM STRATEGY

To ensure robust training, we first select a subset of clean target samples based on classifier consistency on predicted labels ($\mathcal{D}_t^{(0)}$). From this, we further retain adversarial counterparts where classifier predictions remain consistent (*i.e.,* double consistency), forming $\mathcal{D}_t^{\mathrm{adv}}$. To incorporate additional training data progressively, we sort the remaining samples in $\mathcal{D}_t^{(0)} \setminus \mathcal{D}_t^{\mathrm{adv}}$ based on classifier prediction confidence. We define confidence using the average classification confidence, where a larger confidence implies higher certainty in prediction. This strategy enables an easy-to-hard curriculum: high-confidence samples are added earlier, while low-confidence ones are introduced later in training. As shown in Fig. 5, the pseudo-label accuracy consistently decreases across lower confidence percentiles, validating that classifier confidence is a reliable signal for building the curriculum. This progressive expansion of training data not only stabilizes learning but also mitigates the impact of noisy pseudo-labels, leading to improved generalization under domain shift

Table 16: Natural accuracy (Nat.) and robust accuracy (PGD, FGSM, AutoAttack) on the VISDA dataset, evaluated under PGD-20 and FGSM attacks with $\epsilon = \frac{2}{255}$. Table A reports results for the Syn $\rightarrow$ Re domain pairs, and Table B reports results for the Re $\rightarrow$ Syn domain pairs.

(A) Syn $\rightarrow$ Re

| Method | Nat. | PGD | FGSM | AA |
|---|---|---|---|---|
| DANN | $67.4 \pm 0.2$ | $0.5 \pm 0.2$ | $1.8 \pm 0.3$ | $0.0 \pm 0.0$ |
| AT (src only) | $19.0 \pm 0.2$ | $18.0 \pm 0.3$ | $27.4 \pm 0.3$ | $17.2 \pm 0.4$ |
| UDA+AT | $69.6 \pm 0.3$ | $58.3 \pm 0.7$ | $63.2 \pm 0.6$ | $57.5 \pm 0.3$ |
| UDA+TRADES | $68.1 \pm 0.7$ | $57.9 \pm 0.5$ | $61.0 \pm 0.6$ | $56.9 \pm 0.5$ |
| ARTUDA | $45.2 \pm 4.8$ | $32.5 \pm 2.7$ | $37.8 \pm 1.3$ | $31.9 \pm 2.6$ |
| SRoUDA | $48.2 \pm 2.7$ | $33.4 \pm 0.7$ | $39.0 \pm 1.0$ | $30.8 \pm 0.7$ |
| DART | $69.5 \pm 0.2$ | $58.0 \pm 0.5$ | $62.1 \pm 0.4$ | $57.5 \pm 0.6$ |
| CAM+SPLR[†] | 72.8 | 65.9 | − | − |
| **Ours** | $\mathbf{75.3 \pm 0.1}$ | $\mathbf{66.6 \pm 0.3}$ | $\mathbf{70.8 \pm 0.2}$ | $\mathbf{65.0 \pm 0.2}$ |

(B) Re $\rightarrow$ Syn

| Method | Nat. | PGD | FGSM | AA |
|---|---|---|---|---|
| DANN | $78.6 \pm 0.9$ | $0.8 \pm 0.1$ | $2.5 \pm 0.2$ | $0.0 \pm 0.0$ |
| AT (src only) | $53.5 \pm 0.8$ | $41.6 \pm 0.4$ | $52.5 \pm 0.3$ | $39.8 \pm 0.4$ |
| UDA+AT | $85.7 \pm 0.2$ | $82.0 \pm 0.2$ | $84.0 \pm 0.2$ | $81.7 \pm 0.2$ |
| UDA+TRADES | $85.1 \pm 0.3$ | $81.5 \pm 0.5$ | $83.5 \pm 0.4$ | $81.2 \pm 0.5$ |
| ARTUDA | $72.5 \pm 2.5$ | $62.6 \pm 0.3$ | $69.3 \pm 0.7$ | $60.6 \pm 0.4$ |
| SRoUDA | $81.2 \pm 1.4$ | $72.9 \pm 1.3$ | $78.5 \pm 0.6$ | $71.7 \pm 1.6$ |
| DART | $87.3 \pm 0.3$ | $85.3 \pm 0.2$ | $86.2 \pm 0.5$ | $85.1 \pm 0.3$ |
| CAM+SPLR[†] | 89.5 | 87.1 | − | − |
| **Ours** | $\mathbf{89.7 \pm 0.1}$ | $\mathbf{88.0 \pm 0.2}$ | $\mathbf{88.8 \pm 0.2}$ | $\mathbf{87.3 \pm 0.2}$ |

Table 17: Natural (Nat.) and Robust accuracy (PGD) of target test data on the PACS dataset, evaluating across three source-target pairs, under a PGD-20 attack with $\epsilon = \frac{8}{255}$.

| S $\rightarrow$ T | Ph $\rightarrow$ Ar | | Ph $\rightarrow$ Ca | | Ph $\rightarrow$ Sk | |
|---|---|---|---|---|---|---|
| **Method** | Nat. | PGD | Nat. | PGD | Nat. | PGD |
| Natural DANN | 89.1±0.2 | 0.0±0.0 | 80.5±0.2 | 1.1±0.5 | 74.0±1.1 | 0.0±0.0 |
| UDA+AT | 33.8±1.2 | 26.9±0.5 | 81.9±0.6 | 65.1±1.5 | 77.0±0.5 | 72.0±0.5 |
| UDA+TRADES | 62.6±3.4 | 29.7±1.0 | 80.9±1.2 | 66.5±1.3 | 77.2±0.5 | 72.5±0.6 |
| ARTUDA | 26.6±0.7 | 23.5±1.0 | 71.1±2.8 | 52.9±1.7 | 63.4±4.9 | 57.1±3.5 |
| SROUDA | 29.0±0.6 | 24.4±0.8 | 81.2±1.4 | 64.1±0.7 | 50.2±3.8 | 41.9±2.7 |
| DART | 52.8±6.0 | 28.1±2.6 | 88.5 ± 0.1 | 68.9±1.6 | 79.9±0.9 | 74.0±0.9 |
| **Ours** | $\mathbf{58.1 \pm 0.1}$ | $\mathbf{34.5 \pm 0.1}$ | $\mathbf{84.0 \pm 0.1}$ | $\mathbf{74.5 \pm 0.1}$ | $\mathbf{81.7 \pm 0.1}$ | $\mathbf{78.7 \pm 0.2}$ |

and adversarial perturbations. Samples selected via double consistency form a reliable subset, exhibiting consistent confidence levels throughout training. We performed the experiment where we removed both the double-consistency filtering and the curriculum mechanism. Without these components, the model is exposed to strong and unreliable adversarial samples from the very beginning, and we observe that both clean and robust target accuracy drop noticeably (see Table 20).

Table 18: Natural (Nat.) and Robust accuracy (PGD) of target test data on the Digit dataset,evaluating across three source-target pairs, under a PGD-20 attack with $\epsilon = \frac{8}{255}$.

| S→T | S→M | | S→U | |
|---|---|---|---|---|
| Method | Nat. | PGD | Nat. | PGD |
| DANN | 75.5±0.2 | 34.9±1.0 | 87.9±0.6 | 33.7±1.4 |
| UDA+AT | 88.0±0.1 | 84.5±0.1 | 95.5±0.4 | 90.5±0.3 |
| UDA+TRADES | 88.2±0.4 | 84.9±0.6 | 95.5±0.2 | 91.4±0.4 |
| ARTUDA | 96.8±0.7 | 94.2±1.2 | 98.5±0.1 | 94.0±0.8 |
| SROUDA | 87.0±0.3 | 80.4±0.4 | 96.5±0.2 | 88.8±1.0 |
| DART | **99.0±0.0** | 97.8±0.2 | **98.8±0.1** | 96.2±0.1 |
| **Ours** | 99.0 ± 0.1 | **97.8 ± 0.1** | 98.6 ± 0.1 | **96.5 ± 0.1** |

Table 19: Natural (Nat.) and robust (PGD) accuracy on source test data, averaged over all source–target domain pairs across OfficeHome, PACS, VisDA, and Digit datasets. Table A reports results for OfficeHome and PACS dataset, while Table B reports results for VisDA and Digit dataset.

Table A: OfficeHome and PACS

| Method | OfficeHome | | PACS | |
|---|---|---|---|---|
| | Nat. | PGD | Nat. | PGD |
| DANN | **71.2±0.2** | 1.2±0.1 | **89.7±0.4** | 12.3±1.4 |
| AT (src only) | 68.9±0.9 | **47.1±0.3** | 82.9±1.6 | 64.6±1.5 |
| UDA+AT | 45.5±0.2 | 28.0±0.1 | 52.2±1.4 | 33.3±1.1 |
| UDA+TRADES | 45.2±0.8 | 29.6±0.3 | 55.2±1.1 | 36.7±0.7 |
| ARTUDA | 67.3±0.3 | 38.4±0.2 | 85.1±1.0 | 47.0±0.2 |
| SROUDA | 46.2±0.2 | 30.4±0.2 | 58.3±1.4 | 33.7±1.5 |
| DART | 65.1±0.7 | 37.6±0.4 | 85.1±0.1 | 51.5±0.6 |
| **Ours** | 70.5±0.1 | **50.4±0.2** | 87.5±0.5 | **62.3±0.3** |

Table B: VisDA and Digit

| Method | VisDA | | Digit | |
|---|---|---|---|---|
| | Nat. | PGD | Nat. | PGD |
| DANN | **88.0±0.7** | 1.9±0.2 | **98.5±0.5** | 81.8±0.3 |
| AT (src only) | 60.7±6.1 | 47.8±4.9 | 98.3±0.1 | 86.5±0.4 |
| UDA+AT | 37.7±2.2 | 28.7±1.2 | 72.8±0.6 | 65.2±0.3 |
| UDA+TRADES | 37.9±0.6 | 28.9±0.5 | 76.9±0.2 | 69.5±0.4 |
| ARTUDA | 81.0±3.1 | 31.5±2.1 | 72.9±0.6 | 67.6±0.5 |
| SROUDA | 35.1±4.0 | 19.7±1.6 | 65.9±0.4 | 61.5±1.1 |
| DART | 80.9±2.0 | 38.6±2.0 | 95.3±0.5 | 92.8±0.4 |
| **Ours** | 85.2±0.1 | **54.9±0.4** | 97.9±0.2 | **94.5±0.3** |

## E.2 IMPACT OF CONFIDENCE THRESHOLDING WITH CURRICULUM STRATEGY

Many self-training methods rely on confidence thresholding to select pseudo-labeled target samples, retaining only those predictions with softmax scores above a fixed threshold $\tau$. However, such approaches introduce a sensitive hyperparameter that often requires labeled target data for tuning. In our approach, we avoid this issue by identifying a reliable subset of double-consistent samples, which exhibit consistent predictions under both clean and adversarial views. The remaining target samples are sorted by confidence and progressively incorporated into training via an easy-to-hard curriculum strategy (see Section 2.3 in the main paper).

Table 20: Effect of progressive adversarial curriculum on Office-Home (Re → Pr).

| Method Variant | Nat. (%) | PGD. (%) |
|---|---|---|
| w/o Double-Consistency + w/o Curriculum | 74.2 | 63.0 |
| **Ours (w/ DC + Curriculum)** | **78.5** | **68.4** |

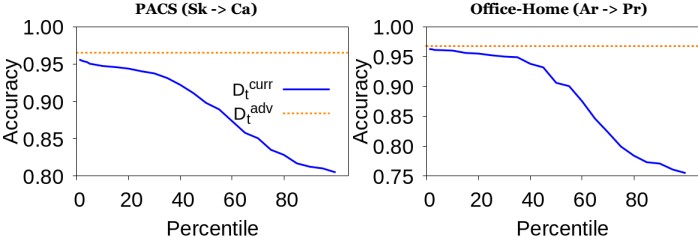

Figure 5: **Pseudo-label accuracy Vs. Confidence Percentile.** We compare the pseudo-label accuracy of two subsets: (i) $\mathcal{D}_t^{adv}$, the double-consistent samples, which exhibit consistently high accuracy throughout training (flat orange line), and (ii) $\mathcal{D}_t^{curr}$, the curriculum-selected samples ranked by decreasing prediction confidence (blue curve). The declining accuracy trend observed in $\mathcal{D}_t^{curr}$ supports the use of an easy-to-hard curriculum strategy, validating that classifier confidence is a reliable proxy for pseudo-label quality.

In a variant of our method, we incorporate softmax-confidence thresholding following Saito et al. (Saito et al., 2017). We follow the same step by selecting target samples where both classifiers agree on the same label. From this subset, we retain only those samples for which at least one classifier produces a softmax maximum above the threshold $\tau$. Curriculum learning is then applied to the resulting high-confidence subset, where samples are sorted by confidence and included in increasing progressive steps. This setup enables us to evaluate the effect of explicit thresholding within the curriculum strategy, while still maintaining a progressive inclusion strategy. Table 21 shows that training performance is sensitive to the choice of confidence threshold $\tau$. The optimal value of $\tau$ is highly dataset-dependent, limiting its applicability across domains.

Table 21: Natural (Nat.) and Robust accuracy (PGD) under confidence-based thresholding ($\tau$) in our method. Results are reported using a PGD-20 attack with $\epsilon = \frac{2}{255}$ on three source–target domain pairs from Office-Home, VisDA, and PACS datasets.

| S → T | Cl → Pr | | Re → Syn | | Ca → Sk | |
|---|---|---|---|---|---|---|
| **Threshold** | Nat. | PGD | Nat. | PGD | Nat. | PGD |
| $\tau = 0.70$ | $59.5 \pm 0.1$ | $49.4 \pm 0.1$ | $84.3 \pm 0.2$ | $82.6 \pm 0.1$ | $82.2 \pm 0.1$ | $79.1 \pm 0.1$ |
| $\tau = 0.60$ | $\mathbf{62.9 \pm 0.1}$ | $\mathbf{52.7 \pm 0.2}$ | $86.7 \pm 0.1$ | $83.4 \pm 0.3$ | $85.7 \pm 0.1$ | $82.2 \pm 0.1$ |
| $\tau = 0.50$ | $61.5 \pm 0.1$ | $50.8 \pm 0.1$ | $88.9 \pm 0.3$ | $87.3 \pm 0.2$ | $86.0 \pm 0.1$ | $83.2 \pm 0.1$ |
| **Ours** | $62.7 \pm 0.2$ | $52.5 \pm 0.3$ | $\mathbf{89.7 \pm 0.1}$ | $\mathbf{88.0 \pm 0.2}$ | $\mathbf{86.2 \pm 0.2}$ | $\mathbf{83.4 \pm 0.3}$ |

## F. MORE RESULTS ON DIFFERENT BACKBONES AND WHITE BOX ATTACKS

We evaluate our proposed method with different backbones including ResNet50, ResNet101 and transformer-based backbones on the VisDA dataset (Table 22), and further assess its robustness under AutoAttack and FGSM on the two domain pair of PACS (Table 23), OfficeHome (Table 24) and VisDA (Table 16) dataset. We also compare the performance of the joint classifier against individual classifiers in Table 25.

Table 22: Natural accuracy (Nat.) and robust accuracy (PGD) on the VISDA dataset, evaluated under a PGD-20 attack with $\epsilon = \frac{2}{255}$, using ResNet-50 ResNet-101 and ViT-B/16 as backbone networks.

| S → T | Syn → Re | | Re → Syn | |
|---|---|---|---|---|
| Method | Nat. | PGD | Nat. | PGD |
| DANN | $67.4 \pm 0.2$ | $0.5 \pm 0.2$ | $78.6 \pm 0.9$ | $0.8 \pm 0.1$ |
| ARTUDA | $45.2 \pm 4.8$ | $32.5 \pm 2.7$ | $72.5 \pm 2.5$ | $62.6 \pm 0.3$ |
| SRoUDA | $48.2 \pm 2.7$ | $33.4 \pm 0.7$ | $81.2 \pm 1.4$ | $72.9 \pm 1.3$ |
| DART | $69.5 \pm 0.2$ | $58.0 \pm 0.5$ | $87.3 \pm 0.3$ | $85.3 \pm 0.2$ |
| CAM+SPLR[†] | 72.8 | 65.9 | 89.5 | 87.1 |
| **Ours(ResNet-50)** | $75.3 \pm 0.1$ | $66.6 \pm 0.2$ | $89.7 \pm 0.1$ | $88.0 \pm 0.2$ |
| **Ours(ResNet-101)** | $78.9 \pm 0.1$ | $69.5 \pm 0.3$ | $92.4 \pm 0.1$ | $89.3 \pm 0.1$ |
| **Ours(ViT-B/16)** | $\mathbf{81.5 \pm 0.1}$ | $\mathbf{74.5 \pm 0.2}$ | $\mathbf{94.6 \pm 0.3}$ | $\mathbf{91.3 \pm 0.2}$ |

Table 23: Natural and robust accuracy (Fast Gradient Sign Method, Projected Gradient Descent, and AutoAttack) evaluated using different methods and backbone networks (Resnet50, Resnet101. ViT-B/16) on Product→Realworld and Product→Art.

| Backbone | Method | Pr→Re | | | | Pr→Ar | | | |
|---|---|---|---|---|---|---|---|---|---|
| | | Nat. | FGSM | PGD | AA | Nat. | FGSM | PGD | AA |
| ResNet-50 | DANN | 60.0±0.6 | 12.2±0.4 | 0.3±0.1 | 0.0±0.0 | 49.1±0.3 | 11.7±0.2 | 0.2±0.1 | 0.0±0.0 |
| | DART | 63.5±0.8 | 54.7±0.2 | 43.6±0.5 | 42.6±0.5 | 43.7±2.5 | 34.5±0.3 | 21.5±0.8 | 20.0±1.0 |
| | Ours | 70.1±0.3 | 62.9±0.1 | 54.7±0.2 | 53.4±0.1 | 50.4±0.2 | 42.7±0.2 | 32.3±0.1 | 30.9±0.3 |
| ResNet-101 | Ours | 75.9±0.3 | 69.2±0.2 | 59.6±0.1 | 56.8±0.1 | 55.7±0.4 | 51.2±0.1 | 37.4±0.1 | 35.2±0.3 |
| DeiT-S/16 | Ours | 76.3±0.1 | 69.5±0.2 | 61.6±0.3 | 59.2±0.3 | 56.5±0.2 | 52.3±0.3 | 40.5±0.3 | 37.4±0.1 |
| DeiT-B/16 | Ours | 80.1±0.2 | 77.3±0.3 | 66.2±0.1 | 64.0±0.1 | 62.4±0.4 | 56.5±0.2 | 44.7±0.2 | 41.5±0.4 |
| ViT-B/16 | Ours | 82.3±0.2 | 79.4±0.1 | 68.6±0.2 | 65.2±0.2 | 64.1±0.1 | 59.3±0.1 | 46.9±0.2 | 43.8±0.3 |

# G. ADDITIONAL ANALYSES AND ABLATION STUDIES

In this section, we provide additional experiments and ablation studies. These analyses validate the design choices in our method, including the the impact of using averaged versus independent cross-entropy objectives role of the discrepancy maximization step.

## G1. ANALYSIS OF AVERAGED VS. INDEPENDENT CROSS-ENTROPY OBJECTIVES

In the warm-start training stage, Eq. (2) defines the cross-entropy loss on the averaged probability of the two classifier heads. Although Eq. (2) is upper bounded by the average of the classifier-wise losses in Eq. (3) via Jensen's inequality, minimizing Eq. (2) does not ensure that each classifier minimizes its own loss. Because the logarithm operates on the averaged probability, the gradients received by each classifier depend on the combined output of both heads rather than on its own prediction. As a result, when one classifier becomes confident while the other is still making mistakes, the confident head dominates the average and masks the errors of the weaker one. The weaker classifier thus receives very small corrective gradients, leading to unbalanced optimization where one head improves disproportionately while the other stagnates.

To address this instability, we adopt the surrogate objective in Eq. (3), which applies cross-entropy independently to each classifier on both clean and adversarial source samples before averaging the resulting losses. This decoupling ensures that each classifier receives a clean gradient signal based solely on its own prediction error. Any disagreement with the ground-truth label immediately increases that classifier's individual loss,

Table 24: Natural and robust accuracy (Fast Gradient Sign Method, Projected Gradient Descent, and AutoAttack) evaluated using different methods and backbone networks (Resnet50, Resnet101. ViT-B/16) on Clipart→Product and Sketch→Realworld.

| Backbone | Method | Cl→Pr | | | | Sk→Re | | | |
|---|---|---|---|---|---|---|---|---|---|
| | | Nat. | FGSM | PGD | AA | Nat. | FGSM | PGD | AA |
| ResNet-50 | DANN | 47.9±0.8 | 9.4±0.3 | 3.6±1.0 | 1.1±0.3 | 67.4±0.2 | 13.5±0.1 | 0.5±0.2 | 0.0±0.0 |
| | DART | 57.0±0.3 | 51.7±0.1 | 45.5±0.6 | 44.8±0.5 | 69.5±0.2 | 62.4±0.3 | 58.0±0.5 | 55.7±0.1 |
| | Ours | 62.7±0.2 | 57.4±0.1 | 52.5±0.3 | 50.9±0.2 | 75.3±0.1 | 70.9±0.5 | 66.6±0.2 | 65.3±0.1 |
| ResNet-101 | Ours | 67.1±0.2 | 63.3±0.1 | 58.1±0.3 | 56.8±0.1 | 78.9±0.1 | 73.4±0.1 | 69.5±0.2 | 67.5±0.3 |
| DeiT-S/16 | Ours | 68.5±0.3 | 64.7±0.2 | 59.8±0.3 | 57.5±0.1 | 79.3±0.1 | 74.8±0.4 | 70.3±0.2 | 67.9±0.1 |
| DeiT-B/16 | Ours | 73.2±0.2 | 70.1±0.1 | 63.6±0.3 | 61.9±0.1 | 80.4±0.2 | 77.2±0.1 | 71.6±0.3 | 69.8±0.3 |
| ViT-B/16 | Ours | 75.3±0.1 | 72.4±0.3 | 66.1±0.2 | 64.9±0.2 | 82.1±0.3 | 78.5±0.1 | 73.4±0.1 | 71.5±0.2 |

Table 25: Comparison of natural and robust accuracy (PGD) between individual classifier heads and their ensemble at inference.

| Dataset | Head H1 | | Head H2 | | Ensemble (H1+H2) | |
|---|---|---|---|---|---|---|
| | Nat. (%) | PGD (%) | Nat. (%) | PGD (%) | Nat. (%) | PGD (%) |
| Office-Home (Re→Cl) | 59.5 | 54.3 | 59.2 | 54.2 | 59.6 | 54.5 |
| PACS (Ca→Sk) | 86.2 | 83.4 | 86.0 | 83.1 | 86.4 | 83.5 |
| VisDA (Syn→Real) | 75.3 | 66.6 | 75.4 | 66.5 | 75.6 | 66.8 |

and the overall objective decreases only when both classifiers assign high probability to the correct class. This naturally drives the two heads toward agreement and results in more stable warm-start optimization.

We validate these observations empirically in two ways. First, we measure the prediction consistency rate between the two classifier heads when trained with Eq. (2) and Eq. (3). As shown in Table 26, Eq. (3) yields substantially higher agreement on both clean and adversarial source samples, confirming that the decoupled objective promotes classifier-wise consistency.

Second, we compare warm-start performance on the Office-Home (Pr → Re) domain pair. Table 27 shows that directly optimizing Eq. (2) produces lower clean and robust accuracy, whereas Eq. (3) yields significantly better performance, consistent with the theoretical motivation for using the surrogate objective.

Table 26: Prediction consistency rate (%) between the two classifier heads when trained with Eq. (2), and Eq. (3) on VisDA (Syn → Real) source data.

| Syn → Real | Eq. (2) | Eq. (3) |
|---|---|---|
| Clean Source | 88.5 | 98.4 |
| Adversarial Source | 85.2 | 96.6 |

## G2. IMPACT OF DISCREPANCY MAXIMIZATION

In explicit alignment stage, we use two classifiers to capture misalignment at the decision-boundary, which cannot be detected through feature-level divergences methods such as MMD (Gretton et al., 2012), CORAL (Sun & Saenko, 2016b) etc. These metrics constrain only marginal feature distributions and correspond to the $\mathcal{H}$-distance; they may become small even when target features lie near unstable or misaligned decision boundaries. In contrast, the discrepancy between $H_1$ and $H_2$ corresponds to the $\mathcal{H}\Delta\mathcal{H}$-distance in the

Table 27: Comparison of warm-start loss on Office-Home (Pr → Rw).

| Warm start loss | Nat. (%) | PGD (%) |
|---|---|---|
| Eq. (2) | 64.5 | 49.2 |
| Eq. (3) | **70.1** | **54.7** |

Ben-David bound (Ben-David et al., 2010), measuring the maximum disagreement between two hypotheses on the target distribution and therefore providing task-aware information. After warm-up, both classifiers agree on the clean and adversarial source domains, forming a shared robust boundary. The maximization step in Eq. (4) then amplifies disagreement on target samples that violate this source-trained decision structure, explicitly revealing boundary-level misalignment. The subsequent minimization step updates the feature extractor to reduce this disagreement, aligning target features toward regions where both classifiers produce consistent and robust predictions.

First, we verify the role of discrepancy maximization, we conduct an ablation (see Table 28) where we remove the maximize step in Eq. (4) and keep all other components of the pipeline unchanged. The "w/o max" variant therefore only minimizes the discrepancy, while our full model uses the standard procedure. On the Office-Home (Pr → Rw) domain pair, removing the maximize step leads to lower clean and robust target accuracy compared to our full method.

To isolate the benefit of discrepancy-based alignment, we replace Eq. (4) with i) MMD (Gretton et al., 2012), ii) CORAL (Sun & Saenko, 2016b) while keeping the architecture and training pipeline identical. Table 29 shows that feature-level divergences yield substantially lower clean and robust accuracy compared to our discrepancy objective, confirming that boundary-aware maximization is essential for robust target alignment.

Table 28: Effect of discrepancy maximization on Office-home (Pr → Rw).

| Method | Nat. (%) | PGD (%) |
|---|---|---|
| w/o maximize discrepancy | 64.7 | 48.2 |
| Ours | 70.1 | 54.7 |

Table 29: Effect of different alignment objectives on target performance Office-Home ( Ar→Cl).

| Alignment Objective | Nat.(%) | PGD (%) |
|---|---|---|
| MMD | 48.7 | 32.4 |
| CORAL | 50.1 | 33.8 |
| Ours | **55.6** | **47.8** |

### G3. COMPARISON OF EXPLICIT ALIGNMENT WITH MCD (SAITO ET AL., 2018A)

Our explicit alignment stage is conceptually different from MCD (Saito et al., 2018a) method because it operates on a latent space already aligned between clean and adversarial source domains, and preserves a stable multi-source decision boundary during discrepancy maximization. MCD maximizes discrepancy directly on a single clean source and does not model adversarial domains or the clean–adversarial gap. In our framework, discrepancy maximization serves only to reveal misaligned target samples, while the minimization step aligns them to a boundary trained on both clean and adversarial supervision. We use a label-free target consistency rate as a convergence criterion that determines when explicit alignment is sufficiently stable to start implicit-training. MCD neither models adversarial domains nor provides such a progressive alignment schedule. It addresses only domain shifts not any convergence criterion. To verify, we replace our explicit alignment with the original MCD objective results in significantly lower clean and robust accuracy (see Table 30).

Table 30: Comparison of explicit alignment with MCD on Office-Home (Ar → Cl).

| Method Variant | Nat. (%) | PGD (%) |
|---|---|---|
| Proposed Method (w/ MCD) | 43.7 | 34.5 |
| **Full Proposed Method** | **55.6** | **47.8** |

## G4. CONVERGENCE ANALYSIS FOR EXPLICIT ALIGNMENT

In the explicit alignment stage, the discrepancy loss exhibits fluctuations due to the adversarial min–max updates and does not provide a stable convergence signal. Therefore, we introduce the Target Consistency Rate (TCR) which increases smoothly and saturates once clean-target representations become consistent across both classifier heads, making it a reliable and label-free criterion for determining when explicit alignment has converged. Unlike MCD (Saito et al., 2018a), which uses a fixed number of alignment iterations, TCR adaptively determines the stopping point based on classifier agreement. The empirical comparison below shows that TCR-based convergence consistently yields higher clean and robust target accuracy.

Table 31: Comparison of convergence criteria on Office-Home (Ar → Cl).

| Method | Nat. (%) | PGD (%) |
|---|---|---|
| Explicit Alignment (10k iters) | $53.2 \pm 0.0$ | $44.6 \pm 0.0$ |
| Explicit Alignment (12k iters) | $54.7 \pm 0.0$ | $45.4 \pm 0.0$ |
| **TCR-based Convergence (Ours)** | **$55.6 \pm 0.3$** | **$47.8 \pm 0.2$** |

## G6. TRAINING AND INFERENCE COMPLEXITY ANALYSIS

We analyze the computational complexity of the proposed three-stage robust UDA framework. The dominant cost in each stage arises from forward–backward passes through the shared feature extractor, classifier heads and the PGD loops used to generate adversarial examples. Let $N_s$ and $N_t$ denote the number of source and target samples processed per epoch, and $K_s$ and $K_t$ denote the numbers of PGD iterations used for generating adversarial samples in the warm-start and implicit alignment stages, respectively. The cumulative training complexity of the full pipeline is: $\mathcal{O}\big(E_{\text{warm}} N_s(1 + K_s) + E_{\text{exp}} \big(N_s(1 + K_s) + N_t\big) + E_{\text{imp}} \big(N_s(1 + K_s) + N_t(1 + K_t)\big)\big)$. This is upper bounded by: $\mathcal{O}\big(E\left(N_s + N_t\right)\left(1 + K\right)\big)$, where $E = E_{\text{warm}} + E_{\text{exp}} + E_{\text{imp}}$ and $K = \max(K_s, K_t)$. Thus, the asymptotic cost matches standard PGD-based adversarial training (Madry et al., 2017; Zhang et al., 2019a), which also requires $K$ inner-loop gradient steps per batch. Other components used in our method—such as classifier discrepancy computation, classifier agreement checks, the Target Consistency Rate, and curriculum-based ranking—incur negligible overhead, consisting of lightweight tensor operations that do not affect the big-$\mathcal{O}$ order. At inference time, no adversarial generation or iterative optimization is required. The model performs a single forward pass through the backbone and both classifier heads, resulting in inference complexity identical to a standard dual-head classifier. Empirically, the method runs at approximately 3 ms/image on a single RTX-3090 GPU, and a full source-to-target training run requires about 5 hours on the same hardware.

## G7. PRACTICAL UTILITY IN REAL-WORLD ROBUST DOMAIN ADAPTATION

The proposed framework is motivated by real-world scenarios where both domain shift and adversarial vulnerability occur simultaneously. Many deployed vision systems such as those used in autonomous driving, robotics, surveillance, or medical diagnostics, operate in environments that differ substantially from their training conditions while being exposed to potential adversarial or perturbation-based threats.

**Autonomous driving.** Traffic sign recognition models are typically trained on labeled images collected in controlled environments (daylight, clear weather, specific cities). When deployed in new conditions—nighttime, fog, rain, different countries—these models face strong domain shift. At the same time, traffic signs can be intentionally manipulated with adversarial stickers or small perturbations that reliably fool camera-based systems. The proposed method jointly aligns clean and adversarial samples across source and target distributions, enabling the system to adapt to new visual conditions while preserving robustness against adversarial manipulations. Conventional UDA methods handle only domain shift, and standard adversarial training addresses only perturbation robustness; neither can manage both simultaneously.

**Medical imaging.** A second representative scenario occurs in cross-hospital medical image adaptation, where models trained on labeled scans from one institution must generalize to unlabeled scans from others. Scanner differences, patient demographics, and acquisition protocols introduce domain shift, and the imaging process itself often contains noise-like perturbations that can degrade prediction fidelity. By jointly aligning clean and adversarial representations, the proposed method reduces both distributional and perturbation-related discrepancies, making it suitable for safety-critical applications where robustness and transferability are required.

These examples illustrate the practical relevance of robust UDA, where distribution shift and adversarial vulnerability coexist. The proposed framework directly addresses this intersection, which is not handled by prior UDA or adversarial training methods in isolation.

## I. DISCUSSION ON CONTRIBUTIONS

- We propose a novel method for robust UDA by framing the problem from a multi-source and multi-target domain alignment, where clean source, adversarial source, clean target, and adversarial target are treated as distinct domains. This formulation captures the inherent distributional shifts among these domains and facilitates more effective alignment for robust generalization.

- Unlike prior work that treats adaptation and robustness independently, our method integrates both into a unified pipeline. We design a progressive domain alignment strategy that begins with explicit alignment through discrepancy minimization on clean target samples, followed by implicit alignment using classifier agreement on pseudo-labeled adversarial target samples. This gradual alignment stabilizes the adaptation process and progressively improves robustness.

- We propose a principled dual-classifier architecture that explicitly aligns decision boundaries across clean and adversarial domains. By measuring classifier disagreement, the model detects boundary-level misalignment that feature-based divergences cannot capture, and uses this signal to guide stable pseudo-labeling. This design provides a unified and robust supervision mechanism that supports reliable adaptation for both clean and adversarial target samples across domains. This dual-head structure enables both robust domain alignment and trustworthy pseudo-label generation for clean and adversarial target data.

- We validate the effectiveness of our method through comprehensive experiments on four benchmark datasets: PACS, Office-Home, VisDA, and Digit. Our method achieves standard accuracies of 62.0%, 88.4%, 82.5%, and 73.7% and the corresponding robust accuracies under PGD-20 attack with $\epsilon = 2/255$ are 49.4%, 78.3%, 77.3%, and 72.1% on the *Office-Home*, *PACS*, *VisDA*, and *Digit* benchmark datasets, respectively. Our results show consistent improvements in both natural and robust accuracy, and component-wise ablations confirm the contribution of each module.