# OpenReview forum: "Progressive Alignment for Robust Domain Adaptation"
_ICLR.cc/2026/Conference — ICLR 2026 Conference Withdrawn Submission_

### Official Review · Reviewer_epLM · 2025-10-28

**Soundness:** 2
**Presentation:** 2
**Contribution:** 2
**Rating:** 4
**Confidence:** 5

**Summary:**

This paper proposes a novel approach to adversarially robust unsuperivsed domain adaptation, which treats the clean and adversarial samples from source and target data as four distinct domains, and handles the alignment with adversarial source domain, clean target domain, and adversarial target domain successively: (1) warm-start training adopts adversarial training on source data; (2) explicit alignment utilizes MCD for clean target domain; (3) implicit alignment adapts the model to adversarial target domain with a curriculum learning strategy from high-confidence to low-confidence adversarial samples, and uses a double consistency criterion for more reliable pseudo-labels. Experimental results suggest the superiority of the proposed method.

**Strengths:**

1. The proposed progressive learning framework and the implicit alignment method are novel for robust UDA.
2. Figures 3 and 4 provide insights into the training process and demonstrate the contribution of different stages.
3. The effectiveness of the proposed method is validated by extensive experiments.

**Weaknesses:**

1. The presentation of the paper requires serious revision to ensure formalness and clarity.
  - $\mathcal{H}$ is undefined in Line 145.
  - Eq. 9-12 are ambiguous due to the reuse of $\mathbf{Z}$ to represent the outputs for different samples.
  - Incorrect citation styles (should use `\citep` for most cases).
  - Missing citations at Lines 709-710.
  - Missing spaces around periods and parentheses (e.g., Line 344) and other typos.
2. The proposed method uses the Target Consistency Rate to determine the convergence of explicit alignment instead of the discrepancy loss used in the original MCD. The superiority of this convergence condition is not empirically verified.
3. Judging from Algorithm 1 (Sec. C), the adversarial source samples are pre-computed before warm-start and fixed during training, instead of being computed based on the latest model parameters. This can significantly deteriorate the effectiveness of adversarial training. The implicit alignment stage may also suffer from the issue of offline adversarial samples, even though the adversarial test samples are updated every $n$ epochs.
4. It is unclear which baseline results are replicated in this paper and which are quoted from the original papers. For quoted results, the comparison may be unfair because the test target data are split for this paper (Lines 811-815) and could differ from those in previous studies. The consistency in experimental settings of the compared methods requires further clarification.
3. The provided source code (anonymous link at Line 845) is buggy, with undefined variables and indentation errors.

**Questions:**

1. Are the results in Section D.2 produced by training the models with $\epsilon=8/255$?

---

> ### Author Response · Authors · 2025-11-21
>
> We thank the reviewer for their valuable and constructive reviews. We have addressed all questions as thoroughly and carefully as possible and to the best of our understanding, and we have incorporated all suggested improvements in the revised version. We apologize for any lack of clarity or typographical errors in the original submission.
>
>
> W1. Thank you for pointing out these issues. We have made the following revisions:
> i).We now clearly define H at its first usage, resolving the missing definition noted at Line 145. For compactness, we define H as the function whose output is the vertical concatenation of the logits produced by the two classifier heads H1​ and H2.
>
> ii) We corrected Equations (9)–(12) by removing ambiguity in the notation for Z; each output is now explicitly indexed by sample type.
>
> iii) All citation styles have been updated to follow the correct \citep convention.
>
> iv) The missing citations at Lines 709–710 have been added.
>
> v) Typographical issues, including missing spaces around punctuation, parentheses, and equation references, have been corrected throughout the manuscript.
>
>
> W2. In the Explicit alignment stage, we observed that the discrepancy loss does decrease but fluctuates significantly due to the adversarial min-max optimization, therefore in practice, we found that it is difficult to determine an appropriate stopping point based solely on the discrepancy value. This makes discrepancy an unreliable indicator of when clean-target features have aligned with the source domains, as shown in the as shown in Figure MCD of repository (https://github.com/Researcharhieve9/adversarial-vision). To obtain a stable and label-free signal, we use the Target Consistency Rate (TCR), which increases smoothly and saturates once clean-target representations become consistent across both classifiers as shown in Fig 3(b) of the main manuscript. In the original MCD framework, the discrepancy loss is optimized during alignment but the method does not rely on it as an explicit convergence signal; instead, the alignment phase is run for a fixed number of iterations. To verify the effectiveness of TCR, we compare MCD based fixed number of training iterations with TCR-based convergence signal. We observe consistently better clean and robust target accuracy with TCR as compared to MCD based fixed training iteration. To add more clarity in the revised version, we will add both the tables and figures.
>
> Empirical comparison of convergence criteria on Ar → Cl (Office-Home).
>
> ---------------------------------------------------------------------------
> Method                          | Nat Acc (%) |  PGD Acc (%)
> --------------------------------|------------------------|------------------------
> Explicit Align (10k iters, MCD)     | 53.2 ± 0.0             | 44.6 ± 0.0
> Explicit Align (12k iters, MCD)     | 54.7 ± 0.0             | 45.4 ± 0.0
> TCR-based Convergence (Ours) | 55.6 ± 0.3   | 47.8 ± 0.2
> ---------------------------------------------------------------------------
>
> W3.  Our method never uses pre-computed or fixed adversarial samples. The confusion arises from the notation in Algorithm 1, where D_s_adv and D_t_adv may appear to be static adversarial datasets. In practice, both adversarial source and adversarial target samples are generated using the current model parameters (F, H1, H2). For every mini-batch in warm-start and in the later stages, we generate new adversarial examples x_s_hat and x_t_hat based on the model’s latest gradient. This ensures that the adversarial gradients always reflect the current state of the network rather than relying on offline perturbations. We have updated Algorithm 1 in the revised manuscript to explicitly indicate that adversarial samples are generated at each iteration for both source and target batches. To add more clarity in the revised version, will rewrite the relevant part in the paper with proper justification and citation.
>
> W4. All baseline results reported in our paper are taken from DART [1] and CAM+SPLR [2]. DART provides a unified and standardized testbed for robust UDA methods, including an identical PGD attack setup and a fixed target-domain test split across all datasets. To avoid any unfairness and distraction from the baselines, we follow the same protocol and use the same target test split as DART and CAM+SPLR. Therefore, the quoted baseline results are consistent with our experimental setup, and all comparisons remain fair and directly comparable. To add more clarity in the revised version, will rewrite the relevant part in the paper with proper justification and citation.
>
>
> [1] Dart: A principled approach to adversarially robust unsupervised domain adaptation.
>
> [2] Toward improving robustness and accuracy in unsupervised domain adaptation.

---

> ### Author Response · Authors · 2025-11-21
>
> W5 The issues in the shared anonymous code (undefined variables and indentation errors) stem from anonymization-related script cleaning and do not affect the internal codebase used to produce all reported results. We fix these artifacts in the revised submission.
>
> Q1.  The results in Section D.2 are obtained by training the models with epsilon = 2/255. This setting follows the standard PGD configuration used in prior robust UDA works [1][2]. Using epsilon = 2/255 during training provides a good balance between robustness and stable feature alignment. The stronger perturbations such as epsilon = 8/255 are known to destabilize the domain-alignment process and significantly degrade the quality of pseudo-labels. Alike [1][2], for evaluation, we report the performance under a stronger attack of epsilon = 8/255 to assess how well the trained model generalizes to more severe perturbations. The evaluation with epsilon = 8/255 is therefore intended to test robustness under a challenging threat model, rather than to reflect the training setup. To add more clarity in the revised version, will rewrite the relevant part in the paper with proper justification and citation.
>
> [1] Dart: A principled approach to adversarially robust unsupervised domain adaptation.
>
> [2] Toward improving robustness and accuracy in unsupervised domain adaptation.

---

### Official Review · Reviewer_LyU6 · 2025-10-28

**Soundness:** 2
**Presentation:** 3
**Contribution:** 2
**Rating:** 4
**Confidence:** 3

**Summary:**

This paper addresses adversarial robustness in the context of unsupervised domain adaptation (UDA). The authors propose a robust UDA framework that treats adversarial examples as samples from distinct source and target domains separate from the clean data. This formulation effectively converts the standard UDA problem into a multi-source, multi-target domain alignment setting. A progressive alignment strategy is then introduced to train the model to align these domains sequentially.

**Strengths:**

The paper proposes an interesting and novel perspective on integrating adversarial robustness and domain adaptation. By incorporating adversarial examples directly into the UDA framework, the method unifies what is often a two-stage training process (UDA followed by adversarial training) into a single, coherent pipeline.

**Weaknesses:**

Further clarifications needed:

- In Line 139, adversarial examples are introduced as belonging to additional domains. However, adversarial perturbations are typically generated with respect to a given classifier. Could the authors clarify which classifier is used to generate these adversarial examples within the “adversarial source” and “adversarial target” domains?
- As the paper studies UDA under adversarial perturbations, the precise **learning objective** and **evaluation metrics** need to be formally defined. What exactly is the target performance criterion (e.g., robust accuracy on target domain under specific attack)? Is the clean accuracy a part of performance criterion as well?
- Regarding loss functions (2) and (3), please elaborate on the challenges of directly optimizing Equation (2). Explaining this difficulty would clarify the motivation for adopting the surrogate objective in Equation (3).
- The paper maximizes the discrepancy between two classifier heads for unlabeled target data (i.e., objective function (4)). Could the authors explain the intuition behind this design? Why should increasing the discrepancy between classifiers help improve adversarial robustness or facilitate domain alignment?
- Line 244-248. Which model is used to generate adversarial examples? Why are _“weak” adversarial examples_ (whose predictions match the clean examples) retained for training? In what sense are these examples considered _“reliable”_? Reliable with respect to what?
- The proposed progressive training strategy begins with “weak” adversarial examples and gradually introduces “stronger” ones. Could the authors justify why such progressive inclusion is necessary? Providing a reasonable explanation would greatly improve the soundness of this design choice.

**Questions:**

See weaknesses.

---

> ### Author Response · Authors · 2025-11-21
>
> We thank the reviewer for their valuable and constructive reviews. We have addressed all questions as thoroughly and carefully as possible and to the best of our understanding, and we have incorporated all suggested improvements in the revised version. We apologize for any lack of clarity or typographical errors in the original submission.
>
>
> W1. Thank you for raising this question. We would like to clarify that the adversarial source and adversarial target are treated as distinct domains, but they are not fixed or pre-computed datasets.  Instead, adversarial source and adversarial target samples are generated during training iteration using PGD attack with respect to the current model parameters, i.e., the shared feature extractor F and both classifier heads H1 and H2. We compute the adversarial perturbation using the gradient of the averaged loss over both classifiers: grad_x ( (1/2) ∑_{i=1}^2 L(H_i(F(x)), y) ), and use this gradient within PGD to generate the adversarial examples x̂_s and x̂_t. This ensures that perturbations reflect the current decision boundaries of the entire model and remain synchronized with the evolving network during training.
>
> W2. Since our objective includes both domain adaptation and adversarial robustness, we evaluate our method using two primary metrics on the target domain: (i) clean target accuracy (\%), and (ii) robust target accuracy (\%) under a PGD-20 attack with epsilon = 2/255, which captures robustness to L_infinity adversarial perturbations. Our method jointly (i) adapts the model to the unlabeled target domain and (ii) ensures adversarial robustness of the target domain. The warm-start stage minimizes the clean and adversarial source loss defined in Eq. 3: L_warm = E_{(x_s, y_s) ∈ D_s^cln} [ (1/2) ∑{i=1}^2 ( -log σ(Z_s^cln[i])[y_s] - log σ(Z_s^adv[i])[y_s] ) ]. The explicit alignment stage uses the classifier discrepancy (Eqs. 4–5): L_exp = E{x_t ∈ D_t^cln} D(p1(x_t), p2(x_t)), with the adversarial updates max_{H1, H2} L_exp followed by min_{F} L_exp. The implicit alignment stage minimizes the robust pseudo-label loss (Eq. 12): L_imp = E_{(x_t, ŷ_t) ∈ D_t^adv ∪ D_t^curr} [ (1/2) ∑_{i=1}^2 ( -log σ(Z_t^cln[i])[ŷ_t] - log σ(Z_t^adv[i])[ŷ_t] ) ]. The complete optimization sequence is: minimize L_warm → (maximize L_exp w.r.t. H1, H2; minimize L_exp w.r.t. F) → minimize L_imp.  Our target performance criteria consist of two metrics, both formally defined. (i) Clean target accuracy (Nat \%) measures the effectiveness of domain adaptation by evaluating the model on unperturbed target test samples. (ii) Robust target accuracy (PGD \%) measures adversarial robustness and is computed under an ℓ∞ PGD-20 attack with perturbation radius ε = 2/255 on the target test samples. These two metrics jointly reflect our objective: achieving high accuracy on the clean target domain while maintaining robustness against adversarial perturbations. Both metrics are consistently reported across all experiments. To add more clarification, we will revise the manuscript.
>
> W3. Equation (2) uses the cross-entropy of the averaged probability produced by the two classifier heads. We found that it is difficult to optimize in practice. The loss mixes the two classifier outputs inside the log, which means that the gradient received by each classifier depends on both probabilities rather than on its own prediction. When one classifier becomes confident while the other is still making mistakes, the confident classifier dominates the average and the weaker classifier receives very little corrective signal. This leads to unbalanced training behavior, where one head improves disproportionately while the other stagnates. To address this issue, we adopt the surrogate loss in Equation (3), which averages the two individual cross-entropy terms. This allows each classifier to learn from its own errors, produces clean and independent gradients, and results in more stable optimization. To verify this effect empirically, we compared Eq. (2) and the surrogate Eq. (3) on the Office-Home Pr → Rw task. Using Eq. (2) led to lower clean and robust accuracy, while Eq. (3) yielded higher accuracy, confirming that the surrogate objective leads to more stable optimization.
>
> Comparison of warm-start losses on Office-Home (Pr → Rw).
>
> | Warm Loss | Clean Acc (%) | Robust Acc (%) |
> |-----------|----------------|----------------|
> | Eq. (2)   | 64.5           | 49.2           |
> | Eq. (3)   | 70.1           | 54.7           |

---

> > ### Author Response · Authors · 2025-11-21
> >
> > W4.  Maximizing the discrepancy between the two classifier heads in Eq. (4) serves to expose target samples that lie outside the decision regions learned from the clean and adversarial source domains. After warm-start, both classifiers agree on multi-source features and therefore share a stable boundary; without the maximize step, their predictions on target samples remain nearly identical, yielding near-zero discrepancy and no usable alignment signal. By encouraging disagreement, Eq. (4) amplifies precisely those target samples whose features are misaligned with source distributions. The subsequent minimization step then updates the feature extractor so that target features move into regions where both classifiers already agree under clean+adversarial supervision. This two-phase process simultaneously aligns target representations and transfers robustness, since the target features are pulled toward a boundary that is already stable under adversarial perturbations. Thus, maximizing discrepancy does not improve accuracy by itself but creates the signal necessary for both domain alignment and robustness transfer. To verify the role of discrepancy maximization, we conduct an ablation where we remove the maximize step in Eq. (4) and keep all other components of the pipeline unchanged. The “w/o max” variant therefore only minimizes the discrepancy, while our full model uses the standard  procedure. On the Office-Home Pr → Rw task, removing the maximize step leads to lower clean and robust target accuracy compared to our full method.
> >
> > Table: Effect of discrepancy maximization on Pr -> Rw (Office-Home).
> >
> > | Method                   | Clean Acc (%) | Robust Acc (%) |
> > |--------------------------|---------------|-----------------|
> > | w/o maximize discrepancy | 64.7          | 48.2            |
> > | Ours                     | 70.1          | 54.7            |
> >
> > W5.  Adversarial examples are generated using PGD with respect to the current model parameters (F, H1, H2). The PGD update uses the gradient of the averaged loss from both classifiers, ensuring that each perturbation reflects the joint decision structure of the model. Weak adversarial examples are those whose predicted labels remain unchanged relative to their clean counterparts. These samples remain within a stable neighborhood of the decision boundary and provide low-noise gradients that help to maintain robust and stable optimization. We retain only those samples whose clean and adversarial predictions agree across both classifier heads (double consistency criteria), indicating that their pseudo-labels are reliable and not affected by noise or ambiguity. Here, reliability refers to the pseudo-label correctness. A target sample is reliable when its clean and adversarial versions produce the same predicted class, and both classifier heads assign this class consistently. This agreement ensures that the sample provides trustworthy supervision for both alignment and robustness transfer. We performed an ablation where the double-consistency criterion was removed. In this setting, all target samples with clean head-agreement are used, even when their adversarial counterparts disagree. This introduces noisy adversarial samples into training and leads to lower clean and robust accuracy.
> > Table: Effect of removing double-consistency filtering on Office-Home (Re -> Pr).
> >
> > | Method                     | Clean Acc (%) | Robust Acc (%) |
> > |----------------------------|---------------|-----------------|
> > | w/o double consistency     | 76.3          | 65.9            |
> > | Ours (with DC)             | 78.5          | 68.4            |

---

> > > ### Author Response · Authors · 2025-11-21
> > >
> > > W6.  If strong adversarial examples are applied at the beginning of training, the model will tend to amplify pseudo-label errors, because the model has not yet aligned its decision boundary with the target distribution. To avoid this instability, we begin with “weak’’ adversarial examples that remain consistent with their clean counterparts (i.e., they satisfy our double-consistency criterion), ensuring that early adversarial supervision comes only from highly reliable samples whose pseudo-labels are likely to be correct. As training progresses and the adversarial target features become better aligned with the clean target boundary, the model can safely handle stronger perturbations.
> > > As per our setting, the weak example is required to satisfy the double consistency criterion. These examples must assign the same prediction by both classifier heads on the both clean and adversarial versions of the sample. The strong example receives consistent predictions from both classifier heads only on its clean setting. We select a subset of adversarial examples from the strong example sets using a confidence-based curriculum. This helps in gradually moving from easy, high-confidence samples to harder ones as the training progresses. This progressive inclusion prevents confirmation bias and allows the adversarial target features to align smoothly with the decision boundary learned from the clean target samples. To verify this, we performed a quick experiment where we removed both the double-consistency filtering and the curriculum mechanism. Without these components, the model is exposed to strong and unreliable adversarial samples from the very beginning, and we observe that both clean and robust target accuracy drop noticeably.
> > >
> > > Table: Effect of removing double-consistency filtering and curriculum (Office-Home Re → Pr).
> > >
> > > | Method                                 | Clean Acc (%) | Robust Acc (%) |
> > > |----------------------------------------|---------------|-----------------|
> > > | w/o DC + w/o Curriculum                | 74.2          | 63.0            |
> > > | Ours (with DC + Curriculum)            | 78.5          | 68.4            |

---

> > ### Author Response · Authors · 2025-11-21
> >
> > W2. In the warm-start stage, although clean and adversarial samples use the same cross-entropy form, they do not influence the model similarly. Adversarial examples are generated by a PGD inner maximization that deliberately finds inputs causing the largest loss increase, so the resulting gradients differ fundamentally from those of clean samples. While clean data preserve the natural manifold, adversarial samples penalize sharp, unstable decision boundaries in the local ( \epsilon )-ball. This adversarial optimization enforces local robustness by smoothing the decision boundary and flattening the loss landscape around each input—well-established effects in adversarial training [1][2]. It is this underlying min–max mechanism, rather than the typical shared CE form that produces robust latent features.
> > The Eq. (10) does not make the loss in Eq. (12) vanish, but to ensure that we only trust adversarial samples whose pseudo-label (\hat{y}_t) is reliable under our double-consistency criterion. Importantly, argmax agreement does not imply that the softmax probability for (\hat{y}_t) is close to 1.0. For adversarial examples, the softmax confidence corresponding to (\hat{y}_t) is typically far below 1 (often in the range of 0.4–0.7), because adversarial perturbations substantially reduce prediction confidence even when the top-1 class remains unchanged. Therefore, even when both classifiers predict (\hat{y}_t) as the argmax, the cross-entropy in Eq. (12) remains strictly positive and provides meaningful gradients. This ensures that Eq. (12) does not become trivial, but instead strengthens the clean–adversarial consistency required for robust alignment.
> >
> > W3.  We thank the reviewer for pointing this out and have added a detailed analysis in the revision. The dominant training cost in our method arises from forward–backward passes through the shared backbone and the PGD loops used to generate adversarial examples. Let (N_s) and (N_t) denote the number of source and target samples per epoch, and (K_s, K_t) the numbers of PGD steps. Our three-stage pipeline therefore has cumulative training complexity ( O\big( E_{\text{warm}} N_s (1+K_s) + E_{\text{exp}} (N_s(1+K_s) + N_t) + E_{\text{imp}} (N_s(1+K_s) + N_t(1+K_t)) \big) ), which is upper bounded by ( O\big(E (N_s+N_t)(1+K)\big) ) with ( E = E_{\text{warm}} + E_{\text{exp}} + E_{\text{imp}} ) and ( K = \max(K_s, K_t) ). This matches the asymptotic order of standard PGD-based adversarial training [1][2]. Additional components—classifier discrepancy, agreement checks, consistency rate, and curriculum ranking—incur only lightweight tensor operations and do not affect the big-O complexity. At inference time, our method requires a single backbone forward pass and two small classifier heads, yielding essentially the same cost as a standard dual-head classifier and no iterative adversarial generation. Thus, both training and inference remain practically feasible. On a single RTX-3090 GPU, one full source→target training run requires approximately 5 hours. At inference time, the model performs a single forward pass through the backbone and classifier heads, resulting in ~3 ms/image latency with no additional adversarial procedures.
> >
> > [1] Madry et al., “Towards Deep Learning Models Resistant to Adversarial Attacks,” ICLR 2018.
> >
> > [2] Zhang et al., “TRADES: A Trade-off Between Accuracy and Robustness,” ICML 2019.

---

### Official Review · Reviewer_4gtK · 2025-10-30

**Soundness:** 2
**Presentation:** 3
**Contribution:** 2
**Rating:** 4
**Confidence:** 3

**Summary:**

The paper addresses the robust unsupervised domain adaptation (UDA) problem setting. The paper proposes a method from the perspective of multi-source and multi-target domain adaptation, treating clean and adversarial samples across both source and target as distinct domains. The proposed method leverages progressive domain alignment strategy that explicitly aligns clean target features with multi-source domains through classifier discrepancy minimisation, and implicitly aligns adversarial target features by enforcing classifier agreement on pseudo-labels.

**Strengths:**

1. The paper addresses robust UDA setting which is an important and practical problem setting.
2. The proposed method is well motivated and the overall paper is well written.

**Weaknesses:**

1. Limited Novelty: The methodologies in the two main modules: Explicit Alignment and Implicit Alignment, appear closely related to prior work. In particular, Explicit Alignment is similar to Saito et al. (2018b), and Implicit Alignment resembles Han et al. (2020). Please clarify what is fundamentally new in your formulation and why these differences matter empirically or theoretically.

2. Methodology needs clearer explanation. In the warm-start stage, why is the loss on adversarial samples defined identically to the loss on clean samples? By design, adversarial examples disrupt effective feature learning, optimizing both sets in the same way makes it unclear how the latent features become robust. Likewise, the Implicit Alignment module requires more detail: from Eq. (10), the pseudo-label is the argmax agreed upon by both classifiers. In that case, wouldn’t the loss in Eq. (12), computed between the same pseudo-label and the logits, tend to be trivially small?

3. Computational complexity analysis is missing: As the proposed method contains multiple stages, a detailed analysis of training and inference complexity is necessary to assess practical feasibility.

4. Missing baselines: In Table 1, several standard UDA methods are missing [R1][R2][R3]. It is important to compare the proposed method with these to understand the effectiveness of the proposed method.


[R1] Zhang, Yuchen, et al. "Bridging theory and algorithm for domain adaptation." International conference on machine learning. PMLR, 2019.

[R2] Rangwani, Harsh, et al. "A closer look at smoothness in domain adversarial training." International conference on machine learning. PMLR, 2022.

[R3] Zhang, Xinyu, Meng Kang, and Shuai Lü. "Low category uncertainty and high training potential instance learning for unsupervised domain adaptation." Proceedings of the AAAI conference on artificial intelligence. Vol. 38. No. 15. 2024.

**Questions:**

1. Explain clealry the differences between the proposed modules, Explicit Alignment and Implicit Alignment with prior works Saito et al. (2018b) and Han et al. (2020), respectively.

2. Explain the methodology clearly. Specifically, In the warm-start stage, why is the loss on adversarial samples defined identically to the loss on clean samples? By design, adversarial examples disrupt effective feature learning, optimizing both sets in the same way makes it unclear how the latent features become robust. Likewise, the Implicit Alignment module requires more detail: from Eq. (10), the pseudo-label is the argmax agreed upon by both classifiers. In that case, wouldn’t the loss in Eq. (12), computed between the same pseudo-label and the logits, tend to be trivially small?

3. Provide a detailed computational complexity analysis to assess the practical feasibility of the proposed method.

---

> ### Author Response · Authors · 2025-11-21
>
> We thank the reviewer for their valuable and constructive reviews. We have addressed all questions as thoroughly and carefully as possible and to the best of our understanding, and we have incorporated all suggested improvements in the revised version. We apologize for any lack of clarity or typographical errors in the original submission.
>
> W1/Q1- **{Explicit Alignment and Implicit Alignment, appear closely related to prior work}**
>
> Ans-We appreciate the reviewer’s careful comparison to Saito et al. (2018b) and Han et al. (2020). Our explicit and implicit alignments are built on fundamentally different concepts and operate in a setting to aim to obtain the aligned decision boundary across domains (S_adv, S_cln, T_cln, T_adv) that jointly addresses both domain shift and the clean–adversarial gap that previous works do not address.
> 1. Our Explicit Alignment is not MCD.
>  Our explicit alignment stage is not a direct application of MCD. We use a classifier discrepancy. Our explicit alignment operates in a latent space that has been pre-aligned between clean and adversarial sources. During discrepancy maximization, we preserve a stable multi-source decision space on S_cln and S_adv while aligning T_cln. We further introduce a label-free Target Consistency Rate as a convergence criterion to determine when explicit alignment becomes stable enough to transition to implicit alignment. MCD does not model adversarial domains, preserve robust margins, and offer a progressive alignment schedule. It addresses only domain shifts.
> 2. Our Implicit Alignment is not Han et al. (2020).
> Our implicit alignment stage is also distinct from Han et al. (2020).  It selects small-loss samples within a single noisy-labeled domain and co-teach two classifiers. In contrast, we construct a double-consistent pseudo-label set requiring both classifier agreement and PGD-stability on target samples. We additionally build an adversarial-hardness curriculum that gradually expands the target training set, which is again guided by the label-free consistency rate. Thus, we use a label-free, robustness-based double-consistency criterion, which does not involve per-sample CE loss or memorization effects. This mechanism is tailored to robust UDA and, to our knowledge, has not appeared in prior UDA or noisy-label literature.
>
> W1/Q1-**{Please clarify what is fundamentally new in your formulation and why these differences matter empirically or theoretically.}**
>
> Ans- We formulate robust UDA as a multi-source multi-target alignment problem over four domains (S_cln, S_adv, T_cln, T_adv), and propose a progressive training pipeline that jointly addresses both the domain shift and the clean–adversarial gap. The differences between our method and prior work are substantive rather than cosmetic, as they directly affect both the theoretical guarantees and the practical behavior of the model. Our warm-start stage builds a robust mixture-source hypothesis, which aligns with multi-source generalization theory[1], whereas MCD uses only a single clean source and therefore has no robustness interpretation. Explicit alignment in our pipeline operates on a robust and stable decision boundary, making the discrepancy term d_HΔH(S_alpha, T_cln) far more reliable than in MCD. Our implicit alignment step uses double-consistency ( clean agreement plus adversarial stability) together with a curriculum strategy to align T_adv, which is theoretically justified through the robust-risk decomposition epsilon_Tadv ≤ epsilon_Tcln + Delta_robust [2]. These design choices directly translate into large empirical gains, especially on adversarial target data, demonstrating that our formulation provides benefits that MCD and the method of Han et al. cannot provide. For a fair ablation, we use the original MCD (without warm start) and the original Han mechanisms (without double-consistency, PGD stability, and curriculum). Empirically, these substitutions significantly reduce both clean and adversarial target accuracy, showing that our explicit and implicit alignment modules are necessary for robust UDA. Theoretically, our multi-source multi-target formulation introduces clean–adversarial divergences that prior methods cannot minimize, explaining why our modules are required.
>
> **Quantitative empirical comparison — Office-Home (Art → Clipart).**
>
> | Model Variant                     | Clean Target Acc (%) | Adv Target Acc (%) |
> |-----------------------------------|-----------------------|---------------------|
> | MCD Replacement (Explicit Align)  | 43.7                  | 34.5                |
> | Han Replacement (Implicit Align)  | 45.1                  | 35.2                |
> | Our Full Model                    | 55.6                  | 47.8                |

---

> ### Author Response · Authors · 2025-11-21
>
> W2/Q2. In the warm-start stage, although clean and adversarial samples use the same cross-entropy form, they do not influence the model similarly. Adversarial examples are generated by a PGD inner maximization that deliberately finds inputs causing the largest loss increase, so the resulting gradients differ fundamentally from those of clean samples. While clean data preserve the natural manifold, adversarial samples penalize sharp, unstable decision boundaries in the local ( \epsilon )-ball. This adversarial optimization enforces local robustness by smoothing the decision boundary and flattening the loss landscape around each input—well-established effects in adversarial training [1][2]. It is this underlying min–max mechanism, rather than the typical shared CE form that produces robust latent features.
> The Eq. (10) does not make the loss in Eq. (12) vanish, but to ensure that we only trust adversarial samples whose pseudo-label (\hat{y}_t) is reliable under our double-consistency criterion. Importantly, argmax agreement does not imply that the softmax probability for (\hat{y}_t) is close to 1.0. For adversarial examples, the softmax confidence corresponding to (\hat{y}_t) is typically far below 1 (often in the range of 0.4–0.7), because adversarial perturbations substantially reduce prediction confidence even when the top-1 class remains unchanged. Therefore, even when both classifiers predict (\hat{y}_t) as the argmax, the cross-entropy in Eq. (12) remains strictly positive and provides meaningful gradients. This ensures that Eq. (12) does not become trivial, but instead strengthens the clean–adversarial consistency required for robust alignment.
>
> W3/Q3.  We thank the reviewer for pointing this out and have added a detailed analysis in the revision. The dominant training cost in our method arises from forward–backward passes through the shared backbone and the PGD loops used to generate adversarial examples. Let (N_s) and (N_t) denote the number of source and target samples per epoch, and (K_s, K_t) the numbers of PGD steps. Our three-stage pipeline therefore has cumulative training complexity ( O\big( E_{\text{warm}} N_s (1+K_s) + E_{\text{exp}} (N_s(1+K_s) + N_t) + E_{\text{imp}} (N_s(1+K_s) + N_t(1+K_t)) \big) ), which is upper bounded by ( O\big(E (N_s+N_t)(1+K)\big) ) with ( E = E_{\text{warm}} + E_{\text{exp}} + E_{\text{imp}} ) and ( K = \max(K_s, K_t) ). This matches the asymptotic order of standard PGD-based adversarial training [1][2]. Additional components—classifier discrepancy, agreement checks, consistency rate, and curriculum ranking—incur only lightweight tensor operations and do not affect the big-O complexity. At inference time, our method requires a single backbone forward pass and two small classifier heads, yielding essentially the same cost as a standard dual-head classifier and no iterative adversarial generation. Thus, both training and inference remain practically feasible. On a single RTX-3090 GPU, one full source→target training run requires approximately 5 hours. At inference time, the model performs a single forward pass through the backbone and classifier heads, resulting in ~3 ms/image latency with no additional adversarial procedures.
>
> [1] Madry et al., “Towards Deep Learning Models Resistant to Adversarial Attacks,” ICLR 2018.
> [2] Zhang et al., “TRADES: A Trade-off Between Accuracy and Robustness,” ICML 2019.

---

> > ### Author Response · Authors · 2025-11-21
> >
> > W4. We agree that these methods are historically important and serve as representative domain-shift UDA approaches. While their original papers report results on several domain adaptation datasets, only the Office-Home dataset overlaps with the datasets used in our study. Therefore, we include only their Office-Home results in the revised Table 1 and mark them with an asterisk (*). We emphasize that these methods are designed solely for  domain shift, and do not incorporate adversarial robustness challenges. Consistent with this, they obtain competitive performance on clean Office-Home targets, where the common UDA method collapses under PGD attack. Our method maintains strong performance on both clean and adversarial target data across all datasets. These observations further highlight that standard UDA methods are not equipped to handle adversarial perturbations, underscoring the necessity and effectiveness of our proposed robust UDA framework.
> >
> >
> > **Table1: Natural (Nat.) and Robust (PGD) accuracy across datasets (Office-Home, PACS, VisDA, Digit).**
> >
> > | Method        | Office-Home Nat | Office-Home PGD | PACS Nat | PACS PGD | VisDA Nat | VisDA PGD | Digit Nat | Digit PGD |
> > |---------------|-----------------|------------------|----------|-----------|------------|------------|------------|------------|
> > | MDD*          | 68.1            | -                | -        | -         | -          | -          | -          | -          |
> > | SDAT*         | 72.2            | -                | -        | -         | -          | -          | -          | -          |
> > | LUHP*         | 75.4            | -                | -        | -         | -          | -          | -          | -          |
> > | Natural DANN  | 57.4            | 1.5              | 81.1     | 11.0      | 73.0       | 6.0        | 72.9       | 68.5       |
> > | AT (src only) | 49.7            | 31.2             | 65.7     | 48.2      | 36.2       | 29.8       | 74.3       | 70.0       |
> > | UDA+AT        | 52.1            | 40.5             | 82.0     | 70.0      | 77.7       | 70.3       | 73.5       | 71.5       |
> > | UDA+TRADES    | 53.4            | 41.0             | 82.3     | 71.7      | 76.6       | 70.2       | 73.3       | 71.5       |
> > | ARTUDA        | 70.3            | 37.1             | 74.6     | 67.4      | 58.9       | 56.2       | 72.7       | 72.1       |
> > | SROUDA        | 51.3            | 38.7             | 72.7     | 64.0      | 64.7       | 53.2       | 73.5       | 71.7       |
> > | DART          | 56.4            | 40.7             | 85.5     | 73.3      | 78.4       | 71.7       | 73.5       | 72.0       |
> > | CAM+SPLR †    | 58.7            | 45.9             | 87.4     | 76.7      | 81.1       | 76.5       | -          | -          |
> > | **Ours**      | **62.0**        | **49.4**         | **88.4** | **78.3**  | **82.5**   | **77.3**   | **73.7**   | **72.1**   |

---

### Official Review · Reviewer_ogfg · 2025-10-30

**Soundness:** 3
**Presentation:** 4
**Contribution:** 2
**Rating:** 6
**Confidence:** 4

**Summary:**

The paper aims to address the adversarial robustness problem in an unsupervised domain adaptation setting. The authors argue that solving both the domain shift and adversarial shift problems jointly, rather than treating them as separate (decoupled) problems, is the right approach. To this end, they reframe the task as a multi-source, multi-target domain adaptation problem, which accounts for both the domain shift and the adversarial gap. The proposed method consists of three phases.

First, a model with two classifier heads is trained on labeled source domains using both normal and adversarial examples. Second, the classifiers and feature extractor are trained alternately in an adversarial manner. Finally, a pseudo-labeling process is used to obtain labels, after which adversarial examples from the target domain are generated and explicit domain adaptation is performed.

Multiple experiments are conducted to demonstrate the performance of the proposed approach. For the experiments, the authors assume access to two labeled domains.

**Strengths:**

Overall, this is a clear and easy-to-follow paper. The authors have identified gaps in prior work, reframed the problem and proposed a method to address them. The proposed approach is sufficiently novel and effectively bridges the identified gap.

**Weaknesses:**

The experimental results are somewhat limited. It would be useful to include additional experiments using transformer-based backbones, such as ViT. Moreover, evaluating the method with standard adversarial attacks, such as AutoAttack [1], would strengthen the results, as PGD-based attacks are known to be prone to certain issues.

Another weakness of the work is practical utility. Authors, in the introduction and abstract mention existing works limited in real-world domain, such as
>yet existing methods fall short in real-world scenarios where adversarial attacks threaten model reliability.

Yet, there is no discussion around this point in the paper.

[1] https://robustbench.github.io/

**Questions:**

In Section 2.2, it is unclear how the authors ensure that adversarial training actually contributes to the overall learning.

The authors assume access to multiple labeled source domains for their experiments, whereas most prior works typically consider only a single labeled source domain. It is therefore unclear how a fair comparison with these prior works can be justified.

---

> ### Author Response · Authors · 2025-11-21
>
> We thank the reviewer for their valuable and constructive reviews. We have addressed all questions as thoroughly and carefully as possible and to the best of our understanding, and we have incorporated all suggested improvements in the revised version. We apologize for any lack of clarity or typographical errors in the original submission.
>
>
> W1. Some quantitative results on different backbones and white-box attacks are provided in Appendix F: Additional Results on Different Backbones and White-Box Attacks (Page 21, Line 964). Furthermore, we update Tables 22 and 23 to include transformer-based backbones beyond ViT-B/16 by adding  DeiT-S/16, DeiT-B/16. Table 21 reports results on VisDA using a ViT-B/16 backbone; Table 16 presents robustness on VisDA under AutoAttack; Table 22 provides AutoAttack evaluations on PACS with ViT-B/16 ,DeiT-S/16 DeiT-B/16; and Table 23 reports the corresponding evaluations on Office-Home using the same set of backbones.  Across these settings, our method consistently improves both natural and robust accuracy and maintains strong performance under AutoAttack, demonstrating effective generalization across transformer-based backbones (ViT-B/16, DeiT-S/16, DeiT-B/16).
>
> Table 22A: Robust accuracy under different attacks on PACS (Cl→Pr).
>
> | Backbone   | Nat  | FGSM | PGD  | AA   |
> |----------- |------|------|------|------|
> | ResNet-50  | 62.7 | 57.4 | 52.5 | 50.9 |
> | ResNet-101 | 67.1 | 63.3 | 58.1 | 56.8 |
> | DeiT-S/16  | 68.5 | 64.7 | 59.8 | 57.5 |
> | DeiT-B/16  | 73.2 | 70.1 | 63.6 | 61.5 |
> | ViT-B/16   | 75.3 | 72.4 | 66.1 | 64.9 |
>
> Table 22B: Robust accuracy under different attacks on PACS (Sk→Re).
>
> | Backbone   | Nat  | FGSM | PGD  | AA   |
> |----------- |------|------|------|------|
> | ResNet-50  | 75.3 | 70.9 | 66.6 | 65.3 |
> | ResNet-101 | 78.9 | 73.4 | 69.5 | 67.5 |
> | DeiT-S/16  | 79.3 | 74.8 | 70.3 | 67.9 |
> | DeiT-B/16  | 80.4 | 77.2 | 71.6 | 69.8 |
> | ViT-B/16   | 82.1 | 78.5 | 73.4 | 71.5 |
>
>
> Table 23A: Robust accuracy under different attacks on Office-Home (Pr→Re).
>
> | Backbone     | Nat  | FGSM | PGD  | AA   |
> |--------------|------|------|------|------|
> | ResNet-50    | 70.1 | 62.9 | 54.7 | 53.4 |
> | ResNet-101   | 75.9 | 69.2 | 59.6 | 56.8 |
> | DeiT-S/16    | 76.3 | 69.5 | 61.6 | 59.2 |
> | DeiT-B/16    | 80.1 | 77.3 | 66.2 | 64.0 |
> | ViT-B/16     | 82.3 | 79.4 | 68.6 | 65.2 |
>
> Table 23B: Robust accuracy under different attacks on Office-Home (Pr→Ar).
>
> | Backbone     | Nat  | FGSM | PGD  | AA   |
> |--------------|------|------|------|------|
> | ResNet-50    | 50.4 | 42.7 | 32.3 | 30.9 |
> | ResNet-101   | 55.7 | 51.2 | 37.4 | 35.2 |
> | DeiT-S/16    | 56.5 | 52.3 | 40.5 | 37.4 |
> | DeiT-B/16    | 62.4 | 56.5 | 44.7 | 41.5 |
> | ViT-B/16     | 64.1 | 59.3 | 46.9 | 43.8 |
>
>
> W2.  A practical example where our framework is very useful is traffic sign recognition in autonomous vehicles. These systems depend entirely on camera images, and the models are typically trained on labeled photos captured in ideal conditions such as daytime, clear weather, and familiar environments. When autonomous vehicles are used in a different city, at night, in fog, or in other low-visibility situations, the model encounters a domain shift. At the same time, traffic signs can also be manipulated through adversarial attacks that are intentionally crafted to confuse the camera. Our method tackles both of these challenges together by adapting the model to the new domain while also keeping it robust to adversarial perturbations. In contrast, existing UDA techniques and adversarial training methods handle these issues separately and therefore often fail in such real-world scenarios.
>
> Another real-world scenario arises in medical imaging, where a model trained on labeled scans from one hospital must operate on unlabeled scans from a different hospital with both domain shift (scanner/protocol differences) and adversarial or noise-like perturbations. Our framework’s joint clean+adversarial across domain alignment allows robust transfer in these safety-critical settings, where existing UDA or adversarial training alone is insufficient.

---

> > ### Author Response · Authors · 2025-11-21
> >
> > Q1-A. Our method contains two adversarial components and each contributes to the overall learning in a different way.
> > (i) In Section 2.1, adversarial training contributes through the warm-start phase defined in Eq. (3), where the feature extractor and both classifier heads are jointly optimized on clean and adversarial source data. Adversarial sources are generated based on PGD based adversarial attack. This step establishes a stable multi-source representation and decision boundaries that already encode adversarial robustness. In explicit alignment, the discrepancy maximization–minimization is performed using these same parameters, so all subsequent updates propagate through a model that has already been shaped by adversarial gradients. Even though the discrepancy loss is computed only on clean target data, the optimization operates on weights that incorporate adversarial information, ensuring that the robustness learned during warm start is preserved and transferred throughout the alignment stage. In implicit alignment, we directly incorporate the adversarial target examples into training which aligns adversarial target features with the stable decision boundary obtained from the previous stages which ensure robustness to the target domain.
> > (ii) In Section 2.2, the “adversarial manner” represents the adversarial-game based optimization which contributes in aligning the clean target domain to robust decision boundary obtained from the warm stage. It simply uses adversarial games by using the Min–max over the two classifiers and min over the feature extractor. Therefore, we keep the feature extractor fixed and update the two classifiers so that they disagree as much as possible on a clean target domain. Then we do the opposite: we keep the classifiers fixed and update the feature extractor so that the discrepancy becomes small. This process pulls the clean target features toward the robust decision boundary. Thus, it provides the stable decision boundary for the implicit alignment stage, where we align adversarial target examples.
> >
> >
> > Q2-A.  In general, multi-domain represents the labeled training data coming from multiple distinct source domains, However, we use the terminology in different aspects. In our experiments, we do not use multiple labeled source domains. For each adaptation task, we strictly follow the standard single-source UDA protocol used in prior work (e.g., Art → Clipart in Office-Home). The term “multi-source–multi-target” refers only to treating the clean and adversarial versions of the same labeled source dataset as two distributions for alignment, and analogously for the target domain. This formulation does not require any additional labeled data. Prior UDA methods treat domain adaptation and adversarial training separately and therefore never align the clean–adversarial gap across domains, which leads to misaligned decision boundaries and reduced robustness. In contrast, our progressive alignment aligns all four distributions S_cln, S_adv, T_cln, T_adv while still using exactly the same source labeled dataset (only one) as previous works.  Therefore, the comparison with prior work is justified

---

### Official Review · Reviewer_FyYi · 2025-11-02

**Soundness:** 2
**Presentation:** 2
**Contribution:** 2
**Rating:** 2
**Confidence:** 4

**Summary:**

This paper addresses the critical challenge of adversarial robustness in Unsupervised Domain Adaptation (UDA) by proposing a novel formulation that treats clean and adversarial samples from both source and target domains as four distinct distributions to be aligned. The authors introduce a progressive alignment strategy that first explicitly aligns clean target features with the multi-source domains and then implicitly aligns adversarial target features by enforcing classifier consistency on refined pseudo-labels. Some experiments on four benchmarks validate that the performance of the proposed method outperforms baselines.

**Strengths:**

Pros:
* The authors conducted a bunch of experiments to validate the effectiveness of their method across four benchmarks.

**Weaknesses:**

Cons:
* Please carefully use \citet and \citep. You should only use \citet when the reference is grammatically part of the sentence, usually as the subject.
* In Lines 171-172, why does minimizing the loss in Eq. (2) produce consistent predictions across two classifiers? And since (2) is upper bounded (instead of lower bounded) by the average of individual losses, it doesn’t mean each individual loss is also minimized due to (2).
* What is the motivation for introducing two classifiers, H1 and H2? The authors use the discrepancy between these two classifiers as the divergence between source and target domains. But this doesn’t make sense to me. And why don’t you just use some widely used divergence metric? Because of this, I don’t think the minimax process in Section 2.2 could align the source and target domain.
* Very limited innovation compared to prior work and methods. This method wants to jointly align the source and target, clean and adversarial, while there are many prior studies [1,2,3] proposing similar frameworks and ideas.

[1]: Exploring Adversarially Robust Training for Unsupervised Domain Adaptation. https://arxiv.org/abs/2202.09300

[2]: Adversarially robust unsupervised domain adaptation. https://www.sciencedirect.com/science/article/pii/S000437022500102X

[3]: Adversarial Feature Alignment: Balancing Robustness and Accuracy in Deep Learning via Adversarial Training. https://arxiv.org/abs/2402.12187

**Questions:**

N/A

---

> ### Author Response · Authors · 2025-11-21
>
> We thank the reviewer for their valuable and constructive reviews. We have addressed all questions as thoroughly and carefully as possible and to the best of our understanding, and we have incorporated all suggested improvements in the revised version. We apologize for any lack of clarity or typographical errors in the original submission.
>
> W1.  We thank the reviewer for pointing this out. We have carefully revised all citations to correctly follow the distinction between \citet and \citep accordingly.
>
> W2.  Eq. (2) defines the cross-entropy loss computed on the averaged probability of the two classifiers. By Jensen’s inequality, Eq. (2) is upper bounded by the average of the classifier-wise cross-entropy losses in Eq. (3). Importantly, we do not minimize Eq. (2), because the averaged probability inside the logarithm couples the two classifiers and allows the classifier that predicts with high confidence to lift the average and hide the errors of a weaker one. This causes the weaker classifier to receive very small gradients, making the loss difficult to optimize. Instead, our method directly uses Eq. (3), which applies cross-entropy separately to each classifier on both clean and adversarial source samples before averaging the results. Because each classifier has its own cross-entropy term, any disagreement with the ground-truth label immediately increases that classifier’s individual loss. The overall loss can only decrease when both classifiers assign high probability to the correct class, which naturally drives them toward consistent predictions.
> To empirically validate this behavior, we measure the prediction consistency rate between the two classifiers on clean and adversarial source samples when trained using Eq. (2) and Eq. (3). As shown in Table, Eq. (3) yields substantially higher prediction consistency on both clean and adversarial source data, confirming that the surrogate objective indeed promotes consistent predictions.
>
> **Prediction consistency rate between the two classifier heads on VisDA (Syn → Real)
> using Eq. (2) vs. Eq. (3).**
>
> | Dataset             | Agreement (Eq2 Objective) | Agreement (Eq3 Objective) |
> |---------------------|----------------------------|----------------------------|
> | Clean Source        | 88.5                       | 98.4                       |
> | Adversarial Source  | 85.2                       | 96.6                       |
>
>
> W3. **Motivation for introducing two classifiers, H1 and H2**
>
> A -The motivation for using two classifiers is twofold.
> (i) They help align the clean target distribution with the multi-source distribution by measuring the disagreement between the classifiers on target samples; higher disagreement indicates that the target features lie farther from the combined clean + adversarial source decision boundary, not merely far in feature space. In contrast, feature-level alignment losses can make clean source, adversarial source, and clean target features appear statistically similar, yet the decision boundaries may remain misaligned. When this happens, the same clean target samples that appear aligned in feature space can be pushed across the decision boundary by small adversarial perturbations, revealing that feature alignment alone does not ensure robust alignment. Dual-classifier disagreement directly exposes this boundary-level mismatch, which feature-distribution metrics cannot detect.
>
>
> (ii) When both classifiers agree on the label of a target sample, the prediction is typically reliable because their agreement indicates that the sample lies in a stable region of the shared robust decision boundary. A single classifier cannot detect when its own prediction is unstable or overconfident under domain shift or adversarial perturbations. Likewise, confidence-, entropy-, or feature-distance–based pseudo-labeling cannot determine whether a sample lies close to an unstable or misaligned decision-boundary region, where even small perturbations can easily flip the prediction

---

> ### Author Response · Authors · 2025-11-21
>
> W3. **The authors use the discrepancy between these two classifiers as the divergence between source and target domains. But this doesn’t make sense to me. And why don’t you just use some widely used divergence metric? Because of this, I don’t think the minimax process in Section 2.2 could align the source and target domain**
>
> Ans-The commonly used divergences such as MMD[2] or CORAL[3] compare only the marginal feature distributions and correspond to the H-distance in the Ben-David bound. These metrics do not reflect how predictions behave near the decision boundary and can be small even when target features are mapped to regions that cause incorrect or unstable predictions. The discrepancy between H1​ and H2​ provides exactly the task-aware information needed during adaptation.
> In contrast, the discrepancy between H1 and H2 is not a generic statistical divergence but corresponds to the HΔH-distance in the Ben-David adaptation bound [1], which measures the maximum disagreement between two hypotheses on the target distribution. During warm-up, the two classifiers are trained to agree on both clean and adversarial source domains. Therefore, any disagreement on a clean target sample directly indicates that the target feature is not aligned with the decision regions learned from the sources.
> After warm-up aligns the source domains in the latent space, the maximization step updates H1​ and H2​ to expose target samples that are inconsistent with the source-trained hypotheses, and the minimization step updates the feature extractor to reduce this discrepancy. This alternating process aligns the target features toward the decision structure learned from the sources.
> To validate the effectiveness of our discrepancy-based alignment, we compare it with two common divergence-based metrics, MMD and CORAL, while keeping the architecture, warm-up, and implicit alignment stages unchanged. In the explicit alignment stage, we replace the classifier discrepancy with  MMD or CORAL computed between the multi-source features and the target features. We then evaluate clean and robust target accuracy for each divergence metric. This controlled comparison isolates the effect of the alignment objective and demonstrates that generic divergence-based feature alignment (MMD/CORAL) is less effective than our task-aware discrepancy alignment.
>
> **Table: Effect of different alignment objectives on target performance (Ar → Cl, Office-Home).**
>
> | Alignment Objective    | Clean Acc (%) | Robust Acc (%) |
> |------------------------|----------------|-----------------|
> | MMD                    | 48.7           | 32.4            |
> | CORAL                  | 50.1           | 33.8            |
> | Ours (Discrepancy)     | 55.6           | 47.8            |
>
> [1] Ben-David, S., Blitzer, J., Crammer, K., Pereira, F., Kulesza, A., Perlmutter, B., and Vaughan, J. W. “A Theory of Learning from Different Domains.” Machine Learning, 79(1–2): 151–175, 2010.
>
> [2] B. Sun and K. Saenko.Deep CORAL: Correlation Alignment for Deep Domain Adaptation. In European Conference on Computer Vision (ECCV), 2016.
>
> [3] A. Gretton, K. Borgwardt, M. Rasch, B. Schölkopf, and A. Smola. A Kernel Two-Sample Test. Journal of Machine Learning Research (JMLR), 13(1):723–773, 2012.

---

> ### Author Response · Authors · 2025-11-21
>
> W4. Our innovations and contributions are as follows:
>
> 1.No prior work formulates UDA with adversarial robustness in the multi-source-multi target-domain setting.
>
> 2.This “progressive robust alignment” is new and not present in prior works.
>
> 3.We present a principled architecture for robust unsupervised domain adaptation, instead of applying decoupled or ad-hoc adversarial training to existing UDA methods.
>
> 4.We establish a new paradigm for robust UDA where robustness and domain shift are integrated into a single coherent pipeline, rather than treated as two unrelated training objectives.
>
> While prior works treat domain alignment and adversarial robustness as separate challenges leads to misaligned decision boundary, clean and adversarial version across domains. Therefore, model tends to overfit on both clean and adversarial target domain. Our method is the first to treat clean and adversarial examples as distinct domains across both source and target (S,T), and to formulate the problem from a multi-source multi-target perspective. We explicitly align decision boundaries across these domains and give a principled dual-classifier architecture that uses boundary consistency to drive robust and domain alignment. This yields a Progressive Alignment Strategy that is fundamentally different from prior approaches and achieves substantially better robustness–transfer trade-offs, empirically and consistent with the theoretical intuition behind domain alignment. Method [1] operates in a fully supervised setting and focuses on supervised pre-training of feature extractor to align the clean and adversarial example in a feature space of the  same domain to achieve the balanced standard and robust accuracy. Methods [2] and [3] address robustness in UDA but mainly align adversarial source and target distributions to efficiently transfer robustness from source to target or generate self-supervised adversarial examples for the target domain and use the UDA methods to train. They do not address the clean–adversarial gaps across domains, and their respective misaligned decision boundary which limits the robust domain adaptation. AT: Adversarial Training, DA: Domain Adaptation
>
> **Comparison with related robust UDA and adversarial training methods.**
>
> | Aspect                                  | [1] ARTUDA | [2] ARUDA | [3] AFA | Ours |
> |-----------------------------------------|------------|-----------|---------|-------|
> | Learning Setting                        | Unsupervised | Unsupervised | Supervised | Unsupervised |
> | Core Problem                            | AT not applicable to unlabeled targets | DA discrepancy fails on adversarial samples | Robustness–clean accuracy tradeoff | due to AT, decision boundaries remain misaligned across domains |
> | Approach                                | Self-supervised adversarial target via KL | Adversarial discrepancy metric for robustness transfer | Adversarial supervised contrastive learning | Progressive multi-source/multi-target alignment (S_cln, S_adv, T_cln, T_adv) |
> | Adversarial Target Usage                | Yes (self-supervised) | No | N/A | Yes (PGD-based from pseudo-labels) |
> | Pseudo-Label Reliability                | No | No | N/A | Yes |
> | Boundary Alignment Across Domains       | No | No | N/A | Yes |
> | Progressive Alignment                   | No | No | No | Yes |
> | Multi-Source / Multi-Target Formulation | No | No | No | Yes |
> | Goal                                    | Enable AT on unlabeled target | Transfer source robustness to target | Balance clean & robust accuracy | Align the decision boundary across Clean- adversarials across domains |
>
> [1]: Exploring Adversarially Robust Training for Unsupervised Domain Adaptation.
>
> [2]: Adversarially robust unsupervised domain adaptation.
>
> [3]: Adversarial Feature Alignment: Balancing Robustness and Accuracy in Deep Learning via Adversarial Training.

---

### Note · Authors · 2026-02-16

I have read and agree with the venue's withdrawal policy on behalf of myself and my co-authors.

---

### Meta-Review · Area_Chair_SGXW · 2026-01-07

**Summary:**

This paper aims to address adversarial robustness in the unsupervised domain adaptation setting. However, several issues remain partially unresolved. Most notably, multiple reviewers continue to express concerns about limited novelty. In addition, the experimental evaluation remains limited, with insufficient comparison to prior works on more extensive backbones and attacks. Overall, while the rebuttal improves clarity and completeness, the novelty and evaluation breadth have room for improvement.

**Reviewer Concerns:**

Reviewer FyYi

W1: Why does minimizing the loss in Eq. (2) produce consistent predictions across two classifiers? [addressed by the rebuttal]

W2: Motivation for introducing two classifiers. [addressed by the rebuttal]

W3: Very limited innovation compared to prior work. [in the middle]

R3: The authors articulate how their method differs from prior work, but the claimed innovations remain largely at a conceptual integration level.

---
Reviewer ogfg

W1: Experimental results are somewhat limited in terms of backbones and attacks. [in the middle]

R1: Although the authors evaluate the proposed method on additional backbones and attacks during rebuttal, there is no comparison with prior works.

W2: Practical utility is not discussed. [still outstanding]

R2: The authors do not address the key:  Why existing methods fall short in real-world scenarios, but the proposed method does not.

---
Reviewer 4gtK

W1: Limited novelty. [in the middle]

The authors differentiate their explicit and implicit alignment modules from prior work, but the novelty still largely stems from integrating and extending existing techniques rather than introducing a fundamentally new method.

W2: The method needs a clearer explanation. [addressed by the rebuttal]

W3: Computational complexity analysis is missing. [in the middle]

R3: Although the authors provide a computational complexity analysis during rebuttal, there is no comparison with prior works.

W4: Missing baselines. [addressed by the rebuttal]

---
Reviewer LyU6

W1: Which classifier is used to generate these adversarial examples? [addressed by the rebuttal]

W2: The precise learning objective and evaluation metrics need to be formally defined. [addressed by the rebuttal]

W3: Elaborate on the challenges of directly optimizing Equation (2). [addressed by the rebuttal]

W4: The intuition behind this design. [addressed by the rebuttal]

W5: Which model is used to generate adversarial examples? [addressed by the rebuttal]

W6: Justify why progressive inclusion is necessary. [addressed by the rebuttal]

---
Reviewer epLM

W1: The superiority of the Target Consistency Rate convergence condition is not empirically verified. [addressed by the rebuttal]

W2: Significantly deteriorates the effectiveness of adversarial training

W3: Which baseline results are replicated in this paper? [addressed by the rebuttal]

W4: The provided source code is buggy. [addressed by the rebuttal]

**Reviewer Scores:**

Reviewer FyYi: 2 -> 2 (some concerns are addressed, but the novelty issue is not fully addressed)

Reviewer ogfg: 6 -> 6 (remains weak accept)

Reviewer 4gtK: 4 -> 4 (some concerns are not fully addressed)

Reviewer LyU6: 4 -> 6 (all confusions are clarified)

Reviewer epLM: 4 -> 4 (some concerns are addressed, but the novelty issue is not fully addressed)

Average score: 4.4

---

### Decision · Program_Chairs · 2026-01-26

Reject